# A release of local subunit conformational heterogeneity underlies gating in a muscle nicotinic acetylcholine receptor

Mackenzie J. Thompson [1], Farid Mansoub Bekarkhanechi [1],
Anna Ananchenko[1], Hugues Nury[2] & John E. Baenziger [1]✉

Synaptic receptors respond to neurotransmitters by opening an ion channel across the post-synaptic membrane to elicit a cellular response. Here we use recent *Torpedo* acetylcholine receptor structures and functional measurements to delineate a key feature underlying allosteric communication between the agonist-binding extracellular and channel-gating transmembrane domains. Extensive mutagenesis at this inter-domain interface re-affirms a critical energetically coupled role for the principal α subunit β1-β2 and M2-M3 loops, with agonist binding re-positioning a key β1-β2 glutamate/valine to facilitate the outward motions of a conserved M2-M3 proline to open the channel gate. Notably, the analogous structures in non-α subunits adopt a locally active-like conformation in the apo state even though each L9′ hydrophobic gate residue in each pore-lining M2 α-helix is closed. Agonist binding releases local conformational heterogeneity transitioning all five subunits into a conformationally symmetric open state. A release of conformational heterogeneity provides a framework for understanding allosteric communication in pentameric ligand-gated ion channels.

A central unanswered question in pentameric ligand-gated ion channel (pLGIC) biology is how the binding of a neurotransmitter to its receptor leads to the opening of a transmembrane gate to allow the flow of ions down their electrochemical gradient into the cell. In the prototypic pLGIC, the muscle-type *Torpedo* nicotinic acetylcholine receptor (nAChR), two binding sites for acetylcholine (ACh) are located at the interfaces between each of the two principal α (α$_\gamma$ and α$_\delta$) and the complementary γ or δ subunits in the extracellular domain (ECD), while the ion channel gate is located ~60 Å away along the central pore axis in the transmembrane domain (TMD) (Fig. 1). Coupling of an ACh-induced conformational change in the ECD with the motions of the pore-lining M2 α-helices in the TMD that open the channel gate is facilitated by both the covalent linkage between β-strand 10 (β10) and the first transmembrane α-helix, M1, and the noncovalent interactions between the β1-β2, β6-β7, and β8-β9 loops from the ECD and the M2-M3 loop from the TMD. How changes to

these inter-domain structures propagate a conformational change from the ECD to the TMD, and vice versa, to facilitate channel gating, however, remains unclear.

Increasing structural data highlight the tertiary/quaternary motions that occur upon agonist binding to pLGICs (Fig. 1 and Movie 1)[1–5]. These motions are typically interpreted in terms of a gating mechanism whereby the closing of loop C around the agonist translates into structural changes in both the β1-β2 and β6-β7 loops and the β10-M1 linker that couple with movements of the M2-M3 loop to open the pore-lining M2 α-helix. It has been suggested that this coupling is mediated by alterations in the overall charging pattern across the ECD–TMD interface or by nonspecific bumping of closely apposed domains[6,7]. On the other hand, the pioneering 4 Å resolution cryo-electron microscopy reconstruction of the muscle-type *Torpedo* nAChR (PDB code 2BG9)[8] identified a molecular continuum in the principal α$_\gamma$/α$_\delta$ subunits leading from each agonist binding site to a

[1]Department of Biochemistry, Microbiology, and Immunology, University of Ottawa, Ottawa, ON K1H 8M5, Canada. [2]Université Grenoble Alpes, CNRS, CEA, IBS, F-38000 Grenoble, France. ✉e-mail: John.Baenziger@uottawa.ca

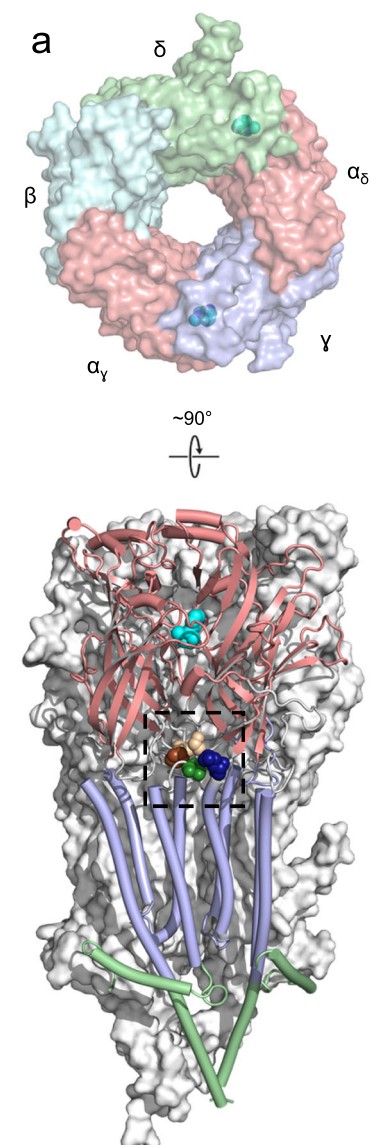

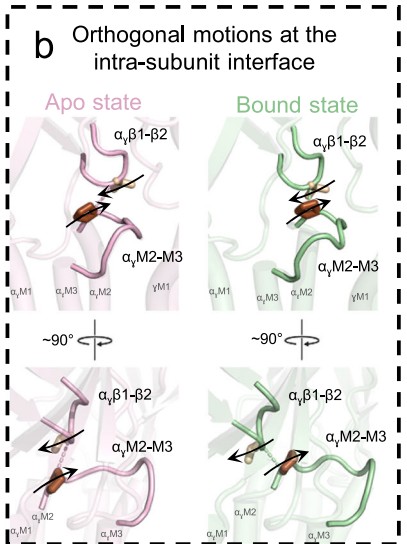

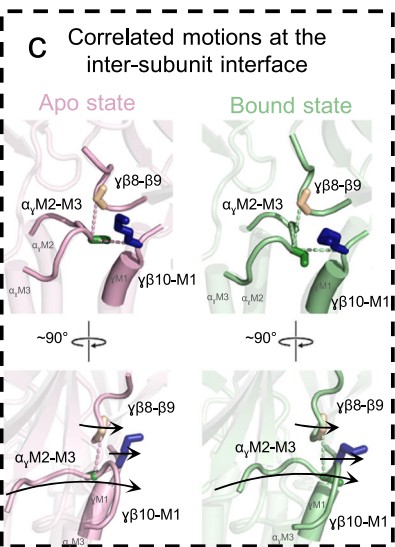

**Fig. 1 | Residues at the ECD – TMD interface allosterically communicate agonist binding into channel gating. a** A top-down view of the *Torpedo* nAChR ECD (PDB: 7QL5) is shown on the top coloured by subunit. A side view of the full receptor is shown on the bottom with the $\alpha_\gamma$ (left) and γ (right) subunits shown as cartoons and coloured according to domain (ECD, salmon; TMD, light blue; ICD, pale green). Bound nicotine is shown as cyan spheres. **b** Zoomed in views of the intra-subunit ECD – TMD interface are shown in apo (pink, PDB:7QKO) and agonist-bound (green, PDB:7QL5) states from two orthogonal views. Arrows depict the roughly orthogonal motions of the loops. **c** Zoomed in views of the inter-subunit ECD – TMD interface are shown in the same orientations as panel b. Arrows depict the concerted motions of the principal α subunit M2-M3 loop and the complementary γ subunit β8-β9 and β10-M1 loops. These motions are visualized in Movie 1.

critical salt bridge between αArg209 at the base of β10 and αGlu45 at the tip of the β1-β2 loop, with αGlu45 and the adjacent αVal46 straddling αPro272 on the M2-M3 loop (Fig. 2a)[8]. Compelling functional data show that these and other residues at the ECD-TMD interface of each muscle α subunit are not only essential for, but couple energetically with each other during channel gating[9–11].

Higher resolution *Torpedo* structures, however, now position αPro272, and other implicated side chains, four residues further along the M2-M3 loop, where they are distant from β1-β2. The structures also show that the tip of β1-β2 not only does not engage tightly with, but moves roughly orthogonal to M2-M3 upon agonist binding (Fig. 2b)[12,13]. Both observations suggest that energetic coupling at the intra-α subunit ECD-TMD interface does not drive channel gating. On the other hand, the extended F loops from the complementary γ/δ subunits rock in upon agonist binding, which causes their membrane juxtaposed β8-β9 loops/β10-M1 linkers to pivot outward (Fig. 1c). Both β8-β9 loops/β10-M1 linkers from γ and δ sandwich the M2-M3 loops from the

adjacent $\alpha_\gamma$ and $\alpha_\delta$ subunits, with these structures moving in concert to open the L9' gate. Both the tight interactions and the concerted agonist-induced motions at these inter-subunit ECD-TMD domain interfaces raise the possibility that it is the γ/δ β8-β9 loops/β10-M1 linkers, not the $\alpha_\gamma/\alpha_\delta$ β1-β2 loops, that primarily couple the movements of the ECD to those of M2-M3 (Movie 1 compares the agonist-induced motions at both the intra- and inter-subunit interfaces of $\alpha_\gamma$ and $\alpha_\gamma$-γ, respectively).

With recent higher resolution structures repositioning many key residues at the ECD-TMD interfaces in the muscle nAChR[12–14], we set out to identify previously unappreciated interactions that regulate allosteric coupling between the ECD and TMD, focusing on both the intra-α subunit and the inter-$\alpha_\gamma/\alpha_\delta$ - γ/δ subunit interfaces. To explore the large number of potential interacting side chains, mitigate the complications arising from integrating mutagenesis data obtained using whole cell versus single channel recordings of muscle nAChRs from different species and with different subunit compositions (fetal

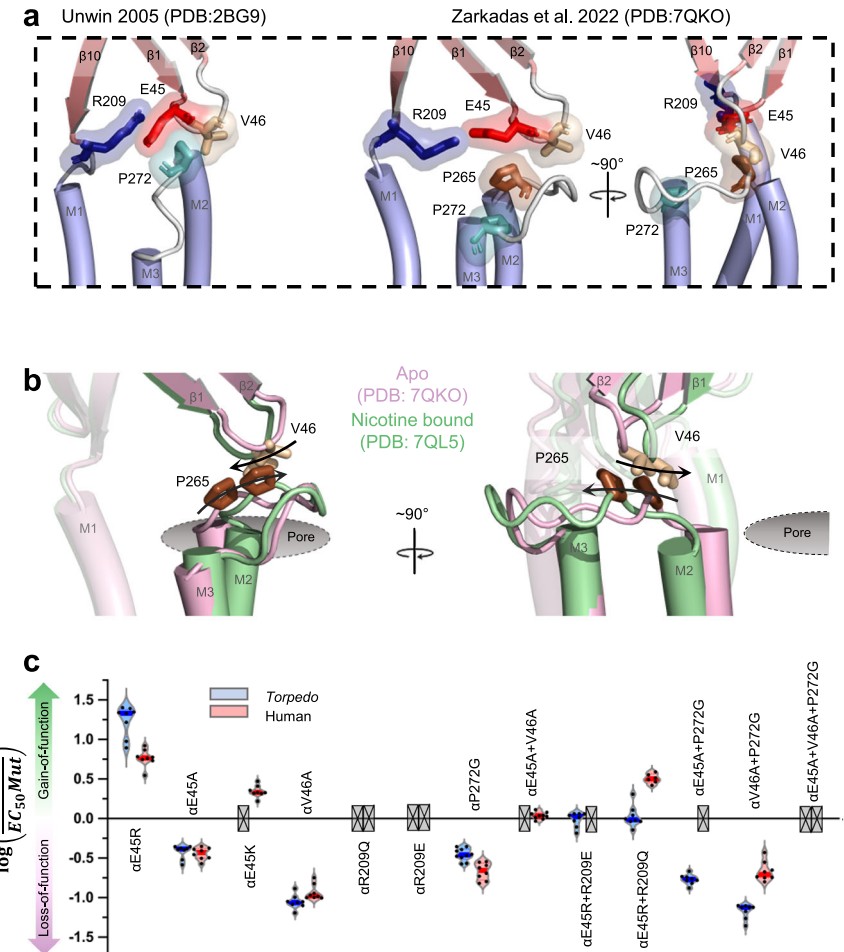

**Fig. 2 | New models reposition key residues at the ECD – TMD interface and show their movements during gating are not coupled. a** A zoomed in view comparing the α subunit ECD – TMD interfaces of the 4.0 Å resolution 2005 (left, PDB: 2BG9) and 2.9 Å resolution 2022 (two orthogonal views on the right, PDB: 7QKO) *Torpedo* nAChR models. Residues in the α subunit β1-β2 and M2-M3 loops that are implicated in channel gating are shown as sticks with transparent surfaces. **b** The movements of the α subunit β1-β2 and M2-M3 loops upon agonist binding are highlighted in the apo (pink, PDB: 7QKO) and nicotine bound (green, PDB: 7QL5) states, with the structures superimposed on their complementary subunits. Arrows on the side chains of αVal46 and αPro265 depict the roughly orthogonal motions of the loops. **c** Violin plots of the measured $EC_{50}$ values for mutations at the intra-α subunit ECD – TMD interfaces of human adult and *Torpedo* nAChRs. Grey crossed boxes indicate mutants that did not produce currents. Exact values can be found in Table S1.

versus adult) expressed in different heterologous systems, and to ensure that our functional data is directly relatable to the *Torpedo* structures, we use a screening mutagenesis approach with each mutation functionally characterized in the *Torpedo* nAChR expressed in *Xenopus* oocytes using two electrode voltage clamp electrophysiology. Despite screening close to 300 mutations, we fail to identify any new interactions that clearly define channel gating. Instead, our integrated structural and functional approach identifies local subunit conformational asymmetry in the apo state at the ECD – TMD interface, with agonist binding releasing this asymmetry so that the nAChR adopts a more symmetric open conformation. We propose that a release of local conformational asymmetry is a key feature underlying allosteric communication at the ECD-TMD interface in the muscle nAChR. Furthermore, the same subunit transitions are observed in other pLGIC structures, albeit with subtle differences expected for homomeric versus heteromeric pLGICs.

## Results

### The intra-α subunit ECD-TMD interface is important to channel function in both human adult and *Torpedo* nAChRs

We first explored the functional role of the intra-α subunit ECD-TMD interface in the context that the compelling functional data suggesting a mechanism involving αArg209, αGlu45, and αVal46 from the ECD

(referred to as the ECD triad) interacting with αPro272 and other residues of M2-M3 were obtained from single-channel recordings of the human adult muscle nAChR[9], while the structural data revealing roughly orthogonal agonist-induced motions of β1-β2 and M2-M3 were obtained using the *Torpedo* nAChR[12]. To definitively rule out the possibility that these apparent discrepancies arise from subtle mechanistic differences between the human adult and *Torpedo* forms, we compared the functional consequences of key mutations in both receptors expressed in frog oocytes (Fig. 1 & S1, Table 1 & S1; representative whole cell traces are presented in Fig. S2). We observed essentially identical functional consequences when each of these residues was mutated suggesting that each residue contributes similarly to channel function in both the adult muscle and *Torpedo* nAChRs. Both data sets highlight an important role for the salt bridge between αArg209 and αGlu45 in that charge reversal or neutralizing mutations of αArg209 lead to no expression while side chain substitutions of αGlu45 lead to large changes in function. On the other hand, the double charge reversal mutations, αE45R + αR209Q and αE45R + αR209E yield close to WT $EC_{50}$ values. Both data sets also show that αVal46 and αPro272 play an important functional role as changing the residues to alanine and glycine, respectively, leads to loss-of-function phenotypes, with αV46A leading to the largest loss-of-function (12-fold) for any single mutation generated in this study. Finally, the αV46A + αP272G and

**Table 1 | Functional effects of mutations to residues at the intra- and inter-subunit ECD-TMD interfaces**

| Mutant[a] | EC$_{50}$ (μM) | Hill slope | $n$ | Fold change |
|---|---|---|---|---|
| WT | 9.40 ± 1.82 | 1.78 ± 0.35 | 87 | |
| Intra-subunit gating interface | | | | |
| αE45R | 0.600 ± 0.316[b] | 1.39 ± 0.21 | 8 | 15.7 gain |
| αE45A | 25.1 ± 5.5[b] | 1.83 ± 0.31 | 8 | 2.67 loss |
| αE45K | NR[c] | | | |
| αV46A | 108 ± 21[b] | 2.00 ± 0.41 | 8 | 11.5 loss |
| αR209Q | NE[d] | | | |
| αR209E | NE[d] | | | |
| αI264A | 28.5 ± 6.3[d] | 1.45 ± 0.19 | 10 | 3.03 loss |
| αP265A | 2.10 ± 0.65[b] | 1.58 ± 0.35 | 9 | 4.47 gain |
| αP265G | 19.8 ± 5.5[b] | 1.85 ± 0.54 | 8 | 2.11 loss |
| αP272G | 27.1 ± 4.9[b] | 1.63 ± 0.15 | 10 | 2.88 loss |
| Inter-subunit gating interface | | | | |
| αS266A | 41.6 ± 7.3[b] | 1.58 ± 0.20 | 8 | 4.42 loss |
| αT267A | 27.4 ± 3.1[b] | 1.57 ± 0.17 | 8 | 2.91 loss |
| αS268A | 2.21 ± 0.33[b] | 1.77 ± 0.28 | 8 | 4.26 gain |
| αS269A | 3.83 ± 0.30 | 1.97 ± 0.47 | 8 | 2.45 gain |
| αS266A + αT267A + αS268A + αS269A | 4.32 ± 1.42 | 2.12 ± 0.26 | 8 | 2.16 gain |
| αS266G + αT267G + αS268G + αS269G | 14.3 ± 2.2 | 1.79 ± 0.13 | 8 | 1.53 loss |
| βE272G + βT273G + βS274G + βL275G | 14.0 ± 1.5 | 1.85 ± 0.18 | 14 | 1.35 loss |
| δE280G + δT281G + δA282G + δL283G | 21.5 ± 3.2[b] | 1.63 ± 0.15 | 7 | 2.28 loss |
| γE275G + γT276G + γS277G + γL278G | 19.7 ± 2.6[b] | 1.51 ± 0.18 | 8 | 2.10 loss |
| αβδγ(STSS → GGGG) | 16.2 ± 3.4 | 2.14 ± 0.47 | 7 | 1.73 loss |
| γG182A | 23.4 ± 2.8[b] | 1.53 ± 0.26 | 8 | 2.49 loss |
| δG188A | 19.3 ± 3.5[b] | 1.59 ± 0.20 | 8 | 2.06 loss |
| γG182A + δG188A | 31.0 ± 2.1[b] | 1.68 ± 0.30 | 9 | 3.33 loss |
| γE183A | 15.9 ± 3.3 | 1.69 ± 0.15 | 8 | 1.69 loss |
| δE189A | 10.2 ± 1.1 | 1.80 ± 0.21 | 8 | 1.09 loss |
| γE183A + δE189A | 16.3 ± 4.0[b] | 1.50 ± 0.51 | 8 | 1.73 loss |
| γK218A | 2.39 ± 0.66[b] | 1.91 ± 0.25 | 8 | 3.94 gain |
| δK224A | 4.24 ± 0.98 | 1.84 ± 0.26 | 9 | 2.20 gain |
| γK218A + δK224A | 0.956 ± 0.434[b] | 2.14 ± 0.57 | 9 | 9.76 gain |
| γP219A | 11.8 ± 1.5 | 2.07 ± 0.56 | 8 | 1.26 loss |
| δP225A | 12.0 ± 1.5 | 1.93 ± 0.59 | 8 | 1.28 loss |
| γP219A + δP225A | 12.0 ± 1.4 | 1.67 ± 0.59 | 8 | 1.28 loss |
| γL220A | 8.83 ± 2.14 | 1.81 ± 0.11 | 8 | 1.06 gain |
| δL226A | 4.48 ± 0.70 | 2.20 ± 0.35 | 8 | 2.08 gain |
| γL220A + δL226A | 6.42 ± 0.81 | 1.87 ± 0.33 | 8 | 1.46 gain |
| γF221A | 17.6 ± 3.0[b] | 1.72 ± 0.28 | 8 | 1.87 loss |
| δF227A | 5.67 ± 0.78 | 2.10 ± 0.71 | 8 | 1.64 gain |
| γF221A + δF227A | 18.0 ± 7.2[b] | 2.04 ± 0.25 | 8 | 1.92 loss |

[a]Measurements performed 2–4 days after cRNA injection (V$_{hold}$ = −60 mV). Error values represented as standard deviation.
[b]$p < 0.001$ relative to WT via one-way ANOVA followed by Dunnet's post hoc test. DF = 343. F$_{EC50}$ = 159.8. F$_{Hill}$ = 3.234. Exact $p$-values can be found in the Source Data file.
[c]No response (NR). No significant agonist-induced current observed up to 4 days after cRNA injection.
[d]No expression (NE). No significant expression or agonist induced current observed up to 4 days after cRNA injection.

the αE45A + αP272G double mutants have less of an effect on the measured EC$_{50}$ values than would be expected if each individual mutation influenced function independently, consistent with the previously reported energetic couplings across this intra-subunit ECD-TMD domain interface (Table 2). Despite structures showing that β1-β2 moves roughly orthogonally to M2-M3 upon agonist binding (Fig. 1b and Movie 1, panel a), our mutagenesis data reaffirm that the ECD triad plays an important role in channel gating (see Fig. S1) and does so, at least in part, by energetically coupling with the M2-M3 loop.

**Side chain interactions at the inter-subunit ECD-TMD interface are not critical for allosteric communication**

We next considered the possibility that agonist-induced motions of the γ/δ β8-β9 loops/β10-M1 linkers play a key role energetic role driving the gating motions of the α$_γ$/α$_δ$ M2-M3 loops. As noted, the apo and agonist bound *Torpedo* structures show that the extended F loops from the complementary γ/δ subunits rock in towards the bound agonist causing their membrane juxtaposed regions, the β8-β9 loops/β10-M1 linkers, to pivot outward, with these outward motions occuring in concert with the outward motions of the α$_γ$/α$_δ$ M2-M3 loops that

**Table 2 | Energetic couplings across the intra- and inter-subunit ECD-TMD interfaces**

| Mutant[a] | EC$_{50}$ (µM) | Hill slope | n | Fold change | Predicted[b] | Ω[c] | ΔΔG (kJ/mol)[d] |
|---|---|---|---|---|---|---|---|
| Intra-subunit ECD-TMD domain interface | | | | | | | |
| αE45R + αR209Q | 9.16 ± 2.47 | 1.92 ± 0.40 | 8 | 1.03 gain | | | |
| αE45R + αR209E | 9.80 ± 2.23 | 1.72 ± 0.09 | 8 | 1.04 loss | | | |
| αE45A + αV46A | NR[f] | | | | 30.6 loss | | |
| αE45A + αP272G | 55.8 ± 8.2[e] | 1.49 ± 0.18 | 8 | 5.93 loss | 7.69 loss | 0.77 | −0.64 ± 0.24 |
| αV46A + αI264A | 143 ± 27[e] | 1.69 ± 0.28 | 8 | 15.2 loss | 34.7 loss | 0.44 | −2.04 ± 0.82 |
| αV46A + αP265A | 32.2 ± 6.0[e] | 2.01 ± 0.26 | 8 | 3.43 loss | 2.56 loss | 1.34 | +0.72 ± 0.33 |
| αV46A + αP265G | 34.2 ± 3.6[e] | 1.99 ± 0.22 | 7 | 3.63 loss | 24.2 loss | 0.15 | −4.69 ± 1.90 |
| αV46A + αP272G | 142 ± 33[e] | 2.29 ± 0.50 | 8 | 15.1 loss | 33.0 loss | 0.46 | −1.94 ± 0.79 |
| αI264A + αP265G | 6.52 ± 1.32 | 1.42 ± 0.33 | 8 | 1.44 gain | 6.38 loss | 0.11 | −5.50 ± 2.48 |
| αI264A + αP272G | 54.5 ± 9.8[e] | 1.59 ± 0.38 | 9 | 5.80 loss | 8.72 loss | 0.66 | −1.01 ± 0.39 |
| αE45A + αV46A + αP272G | NR[f] | | | | 88.1 loss | | |
| αV46A + αI264A + αP265G | 7.80 ± 1.20 | 1.66 ± 0.26 | 8 | 1.21 gain | 73.1 loss | 0.01 | −11.1 ± 5.7 |
| αV46A + αI264A + αP272G | 105 ± 14[e] | 2.11 ± 0.52 | 8 | 11.2 loss | 99.9 loss | 0.11 | −5.43 ± 2.51 |
| Inter-subunit ECD-TMD domain interface | | | | | | | |
| αS266A + γG182A/δG188A | 46.3 ± 6.7[e] | 2.15 ± 0.23 | 8 | 4.93 loss | 22.6 loss | 0.22 | −3.77 ± 1.57 |
| αS266A + γE183A/δE189A | 28.7 ± 4.8[e] | 1.87 ± 0.37 | 8 | 3.06 loss | 8.11 loss | 0.38 | −2.42 ± 1.05 |
| αT267A + γG182A/δG188A | 33.2 ± 3.5[e] | 2.25 ± 0.54 | 8 | 3.53 loss | 14.9 loss | 0.24 | −3.56 ± 1.40 |
| αT267A + γF221A/ δF227A | NR[f] | | | | 3.29 loss | | |
| αS268A + γG182A/δG188A | 11.0 ± 3.1 | 1.82 ± 0.49 | 8 | 1.17 loss | 1.20 loss | 0.98 | −0.05 ± 0.02 |
| αS268A + γK218A/δK224A | 0.486 ± 0.126 | 2.22 ± 0.45 | 8 | 19.4 gain | 37.3 gain | 1.93 | +1.62 ± 0.88 |
| αS268P + γG182A/δG188A | 6.50 ± 2.47 | 1.63 ± 0.55 | 8 | 1.45 gain | 5.24 loss | 0.13 | −5.02 ± 2.09 |
| αS269A + γG182A/δG188A | NR[f] | | | | 2.08 loss | | |
| αS269A + γK218A/δK224A | 0.439 ± 0.126 | 2.03 ± 0.73 | 4 | 21.4 gain | 21.5 gain | 1.00 | 0.00 ± 0.00 |
| αS269A + γL220A/δL226A | NR[f] | | | | 5.49 gain | | |
| αS266A/αT267A + γG182A/δG188A | 31.5 ± 4.6[e] | 2.13 ± 0.32 | 9 | 3.35 loss | 65.7 loss | 0.05 | −7.37 ± 3.49 |
| αS266A/αT267A + γE183A/δE189A | 27.3 ± 9.2[e] | 2.30 ± 0.69 | 8 | 2.90 loss | 23.6 loss | 0.12 | −5.20 ± 2.95 |
| αS268A/αS269A + γG182A/δG188A | 5.05 ± 0.97 | 1.96 ± 0.72 | 8 | 1.86 gain | 2.05 gain | 1.10 | +0.24 ± 0.11 |
| αS268A/αS269A + γK218A/δK224A | 0.176 ± 0.064 | 2.99 ± 0.92[e] | 5 | 53.4 gain | 91.5 gain | 1.71 | +1.33 ± 0.96 |
| αS266G/αT267G/S268G/S269G + γG182A | 20.6 ± 8.2 | 1.65 ± 0.22 | 8 | 2.19 loss | 3.80 loss | 0.58 | −1.36 ± 0.66 |
| αS266G/αT267G/S268G/S269G + δG188A | 22.1 ± 2.4 | 1.87 ± 0.21 | 8 | 2.35 loss | 3.12 loss | 0.75 | −0.70 ± 0.23 |
| αS266G/αT267G/S268G/S269G + γG182A/δG188A | 30.1 ± 5.9[e] | 1.81 ± 0.54 | 8 | 3.20 loss | 7.77 loss | 0.41 | −2.20 ± 0.95 |
| αS266G/αT267G/S268G/S269G + γG182I | 8.16 ± 1.17 | 2.06 ± 0.15 | 8 | 1.15 gain | 2.17 gain | 1.88 | +1.57 ± 0.50 |
| αS266G/αT267G/S268G/S269G + δG188I | 9.49 ± 1.07 | 1.78 ± 0.30 | 8 | 1.01 loss | 1.78 gain | 1.80 | +1.45 ± 0.75 |
| αS266G/αT267G/S268G/S269G + γG182I/δG188I | NR[f] | | | | 5.89 gain | | |

[a]Measurements performed 2–4 days after cRNA injection (V$_{hold}$ = −60 mV). Error values represented as standard deviation.
[b]Predicted fold change if the mutants influenced function independently.
[c]Ω value quantifies the variance from independence (see methods).
[d]An energetic coupling is calculated from EC$_{50}$ values to facilitate comparisons. Energy values provided are not quantitative (see methods and Fig. S1).
[e]$p < 0.001$ relative to WT via one-way ANOVA followed by Dunnet's post hoc test. DF = 270. F$_{EC50}$ = 186.7. F$_{Hill}$ = 4.207. Exact $p$-values can be found in the Source Data file.
[f]No response (NR). No significant agonist induced current observed up to 4 day after cRNA injection.

open the L9' gate (Fig. 3a and Movie 1, panel b). We first probed the functional importance of the γ/δ F loops, which cap around the agonist so that γAsp174/δAsp180 and γGlu176/δGlu182 form hydrogen bonds with the backbone nitrogen of αThr191 (loop C) and the backbone carbonyl of αTyr189 (β9), respectively (Fig. 3b). Capping of loop F is *not* observed in the homomeric α7 nAChR, likely because loop F is too short to close around the bound agonist[5]. We substituted the [172]HIDPED[177]/[178]IIDPEA[183] sequences in the *Torpedo* γ/δ subunits with the corresponding "DISG--" sequence from the α7 loop F to eliminate the noted contacts and thus prevent or minimize this capping motion. The individual substitutions in the γ and δ subunits led to 5- and 2-fold losses of function, respectively, while the simultaneous mutation in both subunits led to a 10-fold loss of function (Table S2). Consistent with the *Torpedo* structures, our functional data suggest that the agonist-induced motions of the two F loops are functionally important.

We next explored whether capping of the F loops contributes to channel gating via interactions between the γ/δ β8-β9 loop/β10-M1 linker and the α$_γ$/α$_δ$ M2-M3 loop. αSer266, αThr267, αSer268, and αSer269 from the α$_γ$/α$_δ$ M2-M3 loop are tightly sandwiched between γGly182/δGly188 and γGlu183/δGlu189 from the complementary γ/δ subunit β8-β9 loop and γLys218/δLys224 from the β10-M1 linker on one side, and γLeu220/δLeu226 and γPhe221/δPhe227 from β10-M1 on the other side (Fig. 3c). We first changed each of the serine/threonine residues in the M2-M3 loop individually to Ala, with αS266A and αT267A leading to 4- and 3-fold losses-in-function, respectively, while αS268A and αS269A led to 4- and 2-fold gains-of-function, respectively (Fig. 3d, Table 1). Remarkably, changing all four residues simultaneously to either Ala or Gly had almost no impact on the measured EC$_{50}$ values (Table 1). For comparison, we repeated the Gly mutations at the aligned positions in the remaining β, γ and δ subunits where M2-

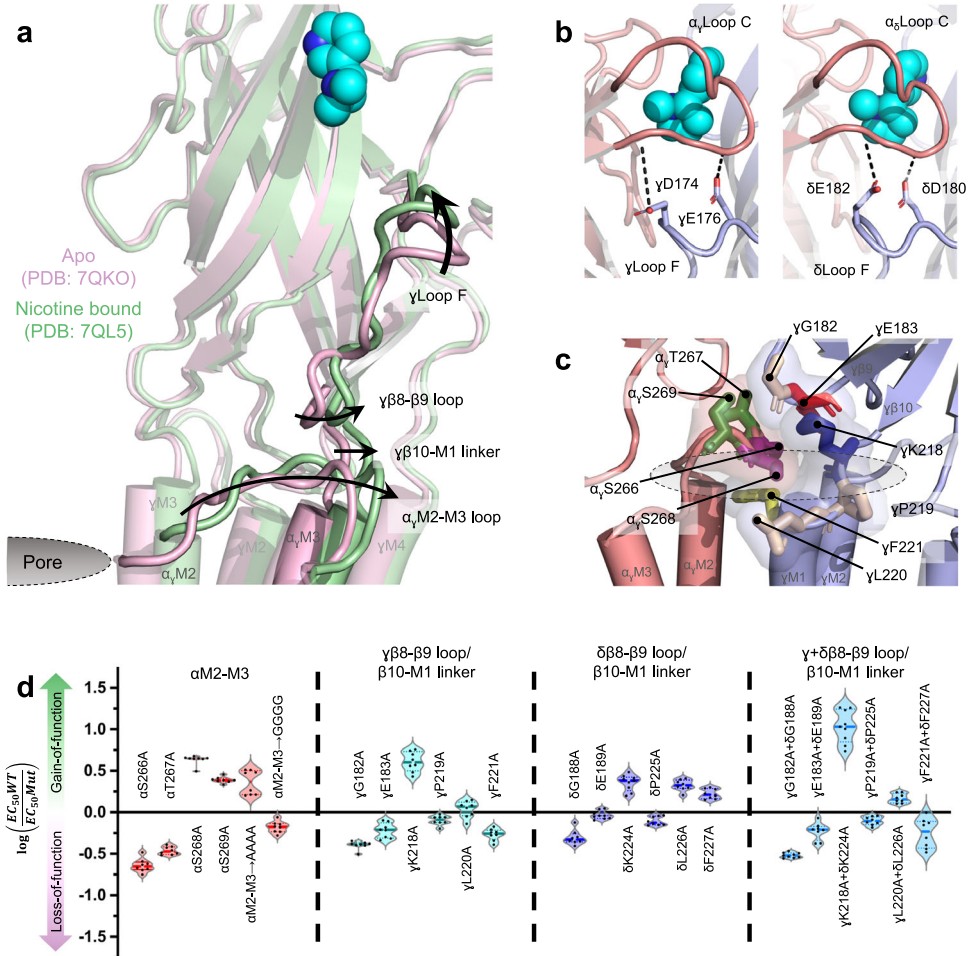

**Fig. 3 | Agonist-induced motions of the complementary γ/δ subunits correlate with the movement of the $α_γ/α_δ$ M2-M3 loops. a** The $α_γ$-γ interface of the *Torpedo* nAChR apo (pink, PDB: 7QKO) and nicotine-bound (green, PDB: 7QL5) states are viewed through the $α_γ$ subunit with the $α_γ$ ECD hidden for clarity. The capping motions of γloop F around the bound agonist (cyan) and the motions of the γβ8-β9, γβ10-M1 and the $α_γ$M2-M3 loops away from the channel pore are depicted by black arrows. **b** Zoomed-in views of the $α_γ$ (left) and $α_δ$ (right) agonist binding sites in the nicotine-bound model, rotated ~90° from the image on the left. The α subunits are

both coloured salmon and the δ/γ subunit coloured light blue. Residues extending from loop F to interact with loop C are shown as sticks with their hydrogen bonds depicted as dashed lines. **c** Residues in the $α_γ$M2-M3 loop, the γβ8-β9 loop, and the γβ10-M1 loop at the ECD – TMD interface are shown as sticks with transparent surfaces highlighting the tight association. **d** Violin plots showing the effects of mutations to residues shown in panel C on channel function are subtle. Exact values can be found in Table 1.

M3 does not translate outwards upon agonist binding and observed the same results. Even when quadruple glycine mutations were generated simultaneously in all five subunits, functional channels with near WT $EC_{50}$ values were observed. The side chains of the four Ser/Thr residues in $α_γ/α_δ$ M2-M3 are not critical to channel function.

Subtle effects on channel function were also observed when mutations were generated to potentially interacting residues on the complementary γ/δ subunits. The γG182A/δG188A and γK218A/δK224A mutations each led to a ~ 3-fold loss or a ~ 4-fold gain of function, respectively, while γE183A/δE189A, γL220A/δL226A and γF221A/δF227A each led to less than two-fold changes in the measured $EC_{50}$ values (Fig. 3d, Table 1). The γK218A + δK224A double mutant did lead to a relatively large ~10-fold gain-of-function, but this appears to be due to the release of a developed steric clash with both αSer268 and αSer269 on M2-M3 (see below). γN181/δN187, γW184, γT185/δE191, γI215, γR217/δR223, γP219, γY222/δY228, γK272 were also changed to non-interacting side chains but these substitutions had minimal functional consequences (Table S2). Although mutations to some of the side chains at the inter-subunit interface are detrimental suggesting functional roles, the relatively small functional consequences for most of the mutations (see Fig. S1 for a discussion of how relative changes in

$EC_{50}$ correlate with changes in channel gating) suggest that the interacting side chains are not critical.

To further test the functional importance of the inter-subunit ECD-TMD domain interface, we cast paired mutations of potentially interacting residues as mutant cycles and found that only two positions in the complementary subunits, γLys218/δLys224 and γGly182/δGly188, couple energetically with residues on $α_γ/α_δ$ M2-M3 (Table 2 and S3). Mutating both lysine (γLys218 and δLys224) and adjacent serine (αSer268 plus αSer269) residues simultaneously to alanine led to a 30-fold gain of function, 3-fold less than would be expected if the mutations influence function independently. This energetic coupling is consistent with a developing steric clash between the side chains that is detrimental to channel activation. Also, changing both γGly182/δGly188 and αSer266/αThr267 residues to alanine led to only a 3.4-fold loss of function despite an expected 66-fold loss of function if the individual mutants influenced function independently. This energetic coupling could be due to the disruption of a hydrogen bond that forms between the backbone carbonyl of γGly182/δGly188 and the backbone amide group of αSer268 upon agonist binding and that contributes energetically to the activated state (agonist binding decreases the hydrogen bond donor - acceptor distance from 3.4 Å to 2.9 Å; Fig. S3)

as the losses of function observed for the γG182A/δG188A mutations no longer affect function on the αS268P background (Fig. S3). Both the γG182A and αS266A + αT267A and the δG188A and αS266A + αT267A triple mutants affect function to a lesser extent than expected given the individual mutations. The triple mutations are also independent on the S268P background. On the other hand, the αS268P alone had no effect on the measured EC$_{50}$ value. These observations are consistent with single channel recordings, which show that mutations to residues in the β8-β9 loop of the human adult muscle nAChR ε subunit typically have minimal effects on the di-liganded gating equilibrium contast[15] and that only detect an energetic coupling across the inter-subunit interface between εGly183 (the *Torpedo* γGly182 equivalent) and αPro265. Although the structural and functional findings suggest that a backbone hydrogen bond between carbonyl of γGly182/δGly188 and the amide of αSer268 at the inter-subunit interface is important to activation, both the relatively small functional consequences of individual mutations and the relatively few energetic couplings between side chains across this interface suggest that side chain interactions at the inter-subunit interface do not play a major role role energetically driving the conformational change that opens the channel gate.

We considered the possibility that non-specific steric interactions couple the motions of γ/δ β8-β9/β10-M1 to those of α$_γ$/α$_δ$ M2-M3[7]. As noted above, replacing all four serine/threonine side chains in α$_γ$/α$_δ$ M2-M3 with four glycine residues to reduce the side chain bulk had essentially no effect on the measured EC$_{50}$ values suggesting that tight steric interactions between α$_γ$/α$_δ$ M2-M3 and γ/δ β8-β9/β10-M1 are not critical. Increasing the bulk and/or charge of residues on the α$_γ$/α$_δ$ M2-M3 loop by replacing all four simultaneously with Asp or Trp led to a complete loss of expression. The quadruple Asn mutant expressed to even higher levels than WT but did not yield agonist-induced currents (Table S4), although molecular dynamics simulations suggest that the loss-of-function results because the mutations energetically stabilize the resting state (Fig. S4). We also noted that the small side chain of γGly182/δGly188 is present in almost all nAChR subtypes and allows the β8-β9 loop to pack tightly against the neighbouring M2-M3 loop. When γGly182 is changed to Ala, Ser or Met there is a subtle loss of function (Table S4). In contrast, the larger branched amino acids Ile, Trp, Glu, Arg, and Val all lead to gains-of-function. To test whether these gains-of-function reflect enhanced steric interactions between γ/δ β8-β9 and α$_γ$/α$_δ$ M2-M3, γG182I/δG188I was superimposed onto the quadruple α$_γ$/α$_δ$ M2-M3 glycine mutant, but this led to a similar gain-of-function to that observed when the mutation was superimposed on the WT background, suggesting that steric interactions between γ/δ β8-β9/β10-M1 and α$_γ$/α$_δ$ M2-M3 do not underlie these functional effects (Table 2).

## Channel opening is accompanied by subunit asymmetric local structural rearrangements at the intra-α subunit ECD-TMD domain interface

Perplexed by the absence of critical side chains/side chain interactions at the inter-subunit ECD-TMD domain interface that drive channel gating, we refocused our attention on the ECD triad of α$_γ$/α$_δ$. We were intrigued by the αV46A mutation, which led to the largest loss-of-function of any single mutation reported in this study. The relatively large negative impact of the αV46A mutation is surprising given that it involves only a reduction in the volume of the side chain. Furthermore, kinetic fitting of single channel measurements suggests the αV46A mutation in the adult muscle nAChR leads to a large ~500-fold decrease in the di-liganded gating equilibrium constant while the αE45A + αV46A double mutation leads to a ~6000-fold reduction in channel gating that almost abolishes the agonist-induced response[9]. The αV46A loss-of-function is driven mainly by a reduction in the channel opening rate constant suggesting that the mutation's main effect is to enhance the energetic stability of the resting or another pre-open closed state. As the reduction in the bulk of the side chain should

reduce, not enhance, local interactions, the large enhancement of the resting state stability likely arises because the bulkier valine side chain locally destabilizes the ECD-TMD domain interface in the closed state, with the smaller alanine side chain reducing this destabilization.

To account for the single channel data, we hypothesized that the energetic coupling between αVal46 and αPro272 established here and elsewhere underlies the local instability created by the bulky valine side chain[9]. Although the original 4 Å resolution 2BG9 structure modeled αGlu45 and αVal46 of β1-β2 straddling αPro272 of M2-M3, higher resolution structures place αPro272 distant from αVal46 and αGlu45, with the latter two residues now projecting towards αPro265[12]. We changed αPro265 to Gly to reduce its steric bulk and this led to a subtle ~2-fold loss of function. When the αV46A mutation was generated on the αP265G background, however, the observed loss of function was reduced from 12-fold to less than 2-fold indicating that there is an energetic connection between these two residues (Table 2). We next hypothesized that the adjacent αIle264, which orients toward M3 and directly contacts αPro272 (Fig. 4a), is a structural link that underlies the energetic coupling between αVal46 and αPro272. Consistent with this hypothesis, the 12-fold loss of function observed for the αV46A mutation was reduced slightly to 5-fold on both the αI264A and the αP272G backgrounds and then reduced further to 2-fold on the αI264A + αP272G background showing that the coupling between αVal46 and αPro272 is dependent upon the intervening αIle264. The available data show that αVal46 couples energetically with αPro265, αIle264, and αPro272, along with other residues in the M2-M3 loop[6–8], with the main point of contact being αPro265.

To better understand how αVal46 couples energetically with αPro265, we re-examined the apo and agonist bound structures and noted that the positions of αGlu45 and αVal46 relative to αPro265 change upon agonist binding (Fig. 4b and Movie 2). In the apo state, both αGlu45 and αVal46 are positioned on the pore distal (i.e., further from the pore axis) side of αPro265, while in the agonist-bound state they are positioned on the pore proximal side (i.e., closer to the pore axis). Agonist binding thus transitions αGlu45 and αVal46 essentially from one side of the bulky αPro265 side chain to the other. This transition likely accounts for the energetic coupling between these residues during channel gating. The transition also suggests an unappreciated role for these three residues in allosteric communication. Specifically, the position of αGlu45/Val46 on the pore-distal side of αPro265 in the apo state may sterically hinder the outward motions of αPro265 that are required to open the channel gate and may thus help restrain M2-M3 in a closed conformation. Agonist-induced motions of β1-β2, and thus αGlu45/Val46, may release the constraints holding the M2-M3 loop in a closed conformation thus facilitating movements outward to open the channel pore.

In this context, we examined the interface between β1-β2 and M2-M3 in each of the non-α subunits, β, γ, and δ, and noted, that each αGlu45/αVal46-equivalent residue is positioned on the pore proximal side of each αPro265-equivalent residue in both apo and agonist-bound states (Fig. 4b). All three non-α subunits thus adopt a locally active-like conformation at this interface in the apo state akin to the agonist-bound conformation in both α subunits. The local subunit conformational heterogeneity rationalizes previous functional data, which highlight an important functional role for residues at the intra-subunit ECD-TMD interface of α, but not non-α subunits[9]. This local conformational heterogeneity, however, does not extend to the remainder of either the ECD or TMD. The apo and agonist-bound ECD conformations of each non-α subunit are both distinct from the apo and agonist-bound α subunit ECD conformations (see Supplemental Discussion of ECD conformational changes)[16]. In contrast, at the hydrophobic gate level, all five M2 α-helices adopt relatively symmetric conformations with both the 9' leucine and 13' valine residues projecting towards the channel pore forming the well-described constriction that prevents cation flux (Fig. 4c). Agonist binding releases

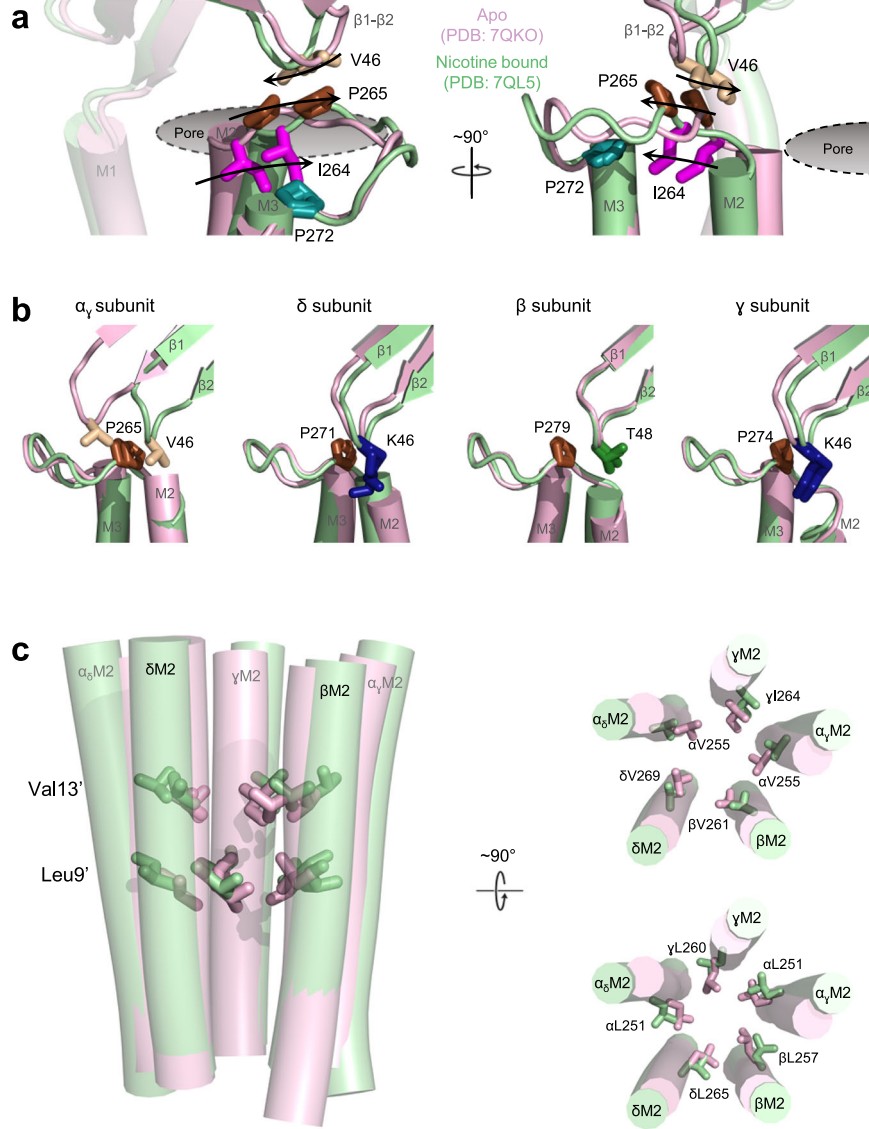

**Fig. 4 | Agonist binding transitions the five intra-subunit ECD − TMD interfaces from a locally asymmetric to a symmetric conformation. a** The ECD − TMD interface of the α_γ subunit is shown in two orthogonal views in apo (pink, PDB: 7QKO) and nicotine bound (green, PDB: 7QL5) states. The roughly orthogonal motions of the β1-β2 and M2-M3 loops create a functional interdependence between αVal46 (tan) and αPro265 (brown) in the apo state with that is propagated to αPro272 (deep teal) via the intervening αIle264 (magenta). **b** The position of αVal46 relative to αPro265, along with their equivalents in the remaining subunits (βLys46, blue; βPro271, brown; δThr48, green; δP279, brown; γLys46, blue; γPro274, brown), are shown for each in both the apo and nicotine bound states. The αVal46 equivalents in the non-α subunits sit on the pore proximal side of the conserved Pro in both states while αVal46 are on the pore distill side in the apo state (asymmetric) and transition to the pore proximal side in the agonist bound state (symmetric). **c** A side view of the pore lining M2 helices are shown on the left for both apo and nicotine bound states with the hydrophobic gate forming V13′ and L9′ residues shown as sticks. Top-down views of the V13′ (top) and L9′ (bottom) residues are shown on the right to highlight the more symmetric conformational changes within the channel pore.

the local conformational heterogeneity at the ECD − TMD interface allowing all five subunits to adopt a more symmetric open state where all five sets of αGlu45/αVal46 equivalent residues are positioned on the pore-proximal side of the corresponding αPro265 equivalent residues and all five 9′ and 13′ side chains at the transmembrane gate have rotated sideways or tilted away from the pore lumen, respectively.

These data suggest an alternative gating mechanism as discussed in more detail below. Instead of driving the allosteric communication between the ECD and TMD, interactions between the two domains, primarily between β1-β2 and M2-M3 of α_γ/α_δ, may help restrain the channel pore in a closed conformation that exhibits local conformational asymmetry, with agonist binding allowing the five subunits to relax into a more symmetric open state. Significantly, essentially the same release of conformational asymmetry is observed in the

heteromeric α1₂β2₂γ2 GABA_AR suggesting that it is a defining feature of gating in all heteromeric pLGICs (see Discussion and See below). Similar motions of β1-β2 relative to M2-M3 have been suggested to underlie gating in the glutamate-activated chloride channel, GluCl[17].

### A tripartite salt bridge is required for whole body motions of the principal α subunit ECD

An agonist-induced whole-body pivoting of each principal α subunit ECD is allosterically linked to the structural changes at the ECD-TMD domain interface that open the channel pore. These concerted movements are likely facilitated by interactions within the ECD β sandwich including a conserved tripartite salt bridge between the αArg209 and the anionic residues αGlu45 (β1-β2), αAsp138 (β6-β7), and αGlu175 (β8-β9). Previous studies have highlighted a critical

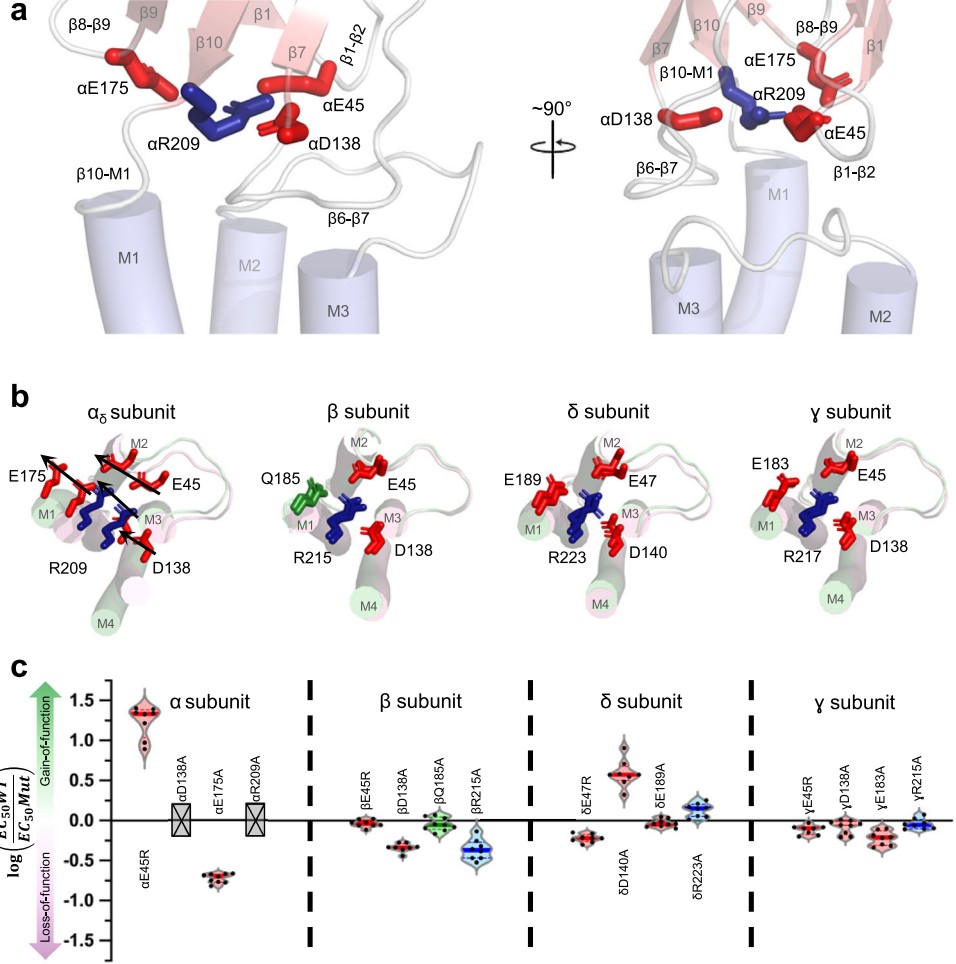

**Fig. 5 | Agonist-induced movements of a tripartite salt bridge at the ECD – TMD interface of each subunit. a** Side views of the ECD - TMD interface of the $\alpha_\gamma$ subunit are shown from two angles with the residues forming the tripartite salt bridges shown as sticks. **b** Top-down views of the apo (pink) and nicotine bound (green) models of the nAChR, aligned by their ECDs, are shown for an α and each non-α subunit with residues composing the tripartite salt bridges shown as sticks. The movements in the α subunit are depicted by arrows while the non-α subunits remain essentially static. **c** Violin plots of the fold changes in EC$_{50}$ values for mutations to each residue in the tripartite salt bridges of each subunit in the *Torpedo* nAChR. Grey crossed boxes indicate mutants that did not produce currents. Exact values can be found in table S5.

functional role for the salt bridge between αArg209 and the anionic residues αGlu45 (β1-β2)[6,9,15,18–22]. In the *Torpedo* nAChR, both charge neutralization and reversal mutants of αArg209 did not express (Fig. 5c, Table S5). Mutations to the coordinated anionic residues were better tolerated than those of αArg209, presumably because they can partially compensate for the loss of each other, but still lead to relatively large changes in EC$_{50}$ values (Table S5). Considerable functional data show that the tripartite salt bridge at the base of the principal subunit ECD is critical to channel function[6,9,15,18,20,21].

In contrast, other than the movements of loop F in γ and δ, the tertiary structures of each non-α subunit ECD is maintained throughout the transition so that each entire subunit moves essentially as a rigid body (Movie 3). Given the lack of whole-body pivoting motions of each non-α subunit ECD, we hypothesized that the tripartite salt bridge in β, γ, and δ would not play a critical role in channel function. Consistent with this hypothesis and in agreement with previously published data[6,9], mutations to the analogous charged residues in these non-α subunits are well tolerated, expressing robustly and with EC$_{50}$ values close to WT (Table S5). For example, charge neutralization of the central arginine of the tripartite salt bridge was tolerated in each of the β, γ, and δ subunits with little effect on channel gating. Furthermore, while the charge reversal R217E mutation in the γ subunit did not express, the same charge reversals had no effect on channel function

in either the β or δ subunits. These findings highlight the importance of electrostatic bridging interactions in the ECD of $\alpha_\gamma/\alpha_\delta$, while suggesting that whole body ECD motions in the β, δ, and γ do not contribute to channel function.

## Discussion

We set out to identify interactions that facilitate allosteric communication between the agonist site and the channel gate in the dual context of new *Torpedo* structures that reposition many key residues at the ECD-TMD domain interface, and a body of diverse mutagenesis data obtained from different labs using muscle nAChRs from different species and with different subunit compositions expressed in different heterologous systems. To mitigate the complications arising from the latter, we used a screening mutagenesis approach with each mutant functionally characterized in *Xenopus* oocytes using two-electrode voltage clamp electrophysiology and focused on mutations in the *Torpedo* nAChR, which are unambiguously relatable to the *Torpedo* structures. A key finding of our data, bolstered by single channel measurements[9,15], is that larger loss-of-function mutations and more extensive energetic couplings occur at the intra-α subunit ECD-TMD interface than at the inter-subunit ECD-TMD interface despite the fact that the interacting residues at the intra-α subunit interface do not engage tightly with each other and move roughly orthogonally to each

other while interacting residues at the inter-subunit interface engage tightly and move in concert upon agonist binding. Even changing all four Ser/Thr residues on $\alpha_\gamma/\alpha_\delta$ M2-M3, which engage with the β8-β9 loops/β10-M1 linkers from γ/δ, to Ala or Gly has little effect on channel function. These contradictory structural and functional observations are not easily reconciled in the context of the mechanisms that are typically used to frame agonist-induced channel gating. In such models, agonist binding leads to movements of the ECD that result in the formation of new or enhanced interactions across the ECD-TMD domain that energetically stabilize the open pore. Although such interactions do occur, such as the formation of a backbone hydrogen bond between the carbonyl of γGly182/δGly188 and the amide group of αSer268, the inability to identify consensus interactions across the ECD-TMD interface that define the open state, despite the extensive mutagenesis studies performed here and elsewhere[6,7], suggests that such open state defining interactions do not exist. Instead, we propose that allosteric communication is governed by conformational restraints imposed primarily at the α subunit ECD-TMD interface, with agonist-binding releasing these restraints allowing the nAChR pentamer to relax from a locally asymmetric to a more symmetric open state. We base this model on the several sets of observations.

First, our and other mutagenesis data highlight the functional importance of residues at the interface between the β1-β2 and M2-M3 loops of the α subunit, notably αGlu45 and αVal46 from β1-β2 and αPro265 from M2-M3, but not the analogous residues in non-α subunits[9]. In fact, it has been suggested that αPro265 plays key role orchestrating the agonist-induced response[3]. Our data also highlight an energetic link between αGlu45/αVal46 and αPro265 that likely stems from the structural rearrangements of these residues upon agonist binding. Specifically, αGlu45/αVal46 are positioned in the apo state on the pore distal of αPro265 where they may sterically prevent the outward motions of M2-M3 that are required to open the channel gate and thus may constrain M2-M3 in a closed conformation. The β8-β9 loop/β10-M1 linker from γ/δ also interact extensively with M2-M3 from $\alpha_\gamma/\alpha_\delta$ in a manner that may restrain M2-M3 from transitioning outward into an open state, although side chain interactions at this interface appear less critical to channel function. Agonist-induced motions of the $\alpha_\gamma/\alpha_\delta$ ECD that move its membrane juxtaposed regions upward and sideways toward the adjacent β/γ subunits, along with motions of the γ/δ β8-β9 loop/β10-M1 linker, may simply release $\alpha_\gamma/\alpha_\delta$ M2-M3 allowing it to translate outward to open the pore, with wetting and de-wetting of the pore possibly playing a significant role establishing the energetics between closed and open states[3,23]. Although there are conflicting reports as to whether the isolated TMD exhibits an energetic preference for closed or open states[24], the hypothesis that the ECD restrains the pore in a closed conformation is consistent with the observation that isolated TMDs of the nAChR and the glycine receptor both form pentamers that not only undergo spontaneous opening/closing transitions, but that have a greater propensity to adopt an open conformation than the intact receptors[25-27]. These biophysical studies led to a similar hypothesis that the ECD restrains the pore in a closed conformation with agonist-binding releasing the restraints so that the TMD rapidly relaxes into its preferred open conformation[28]. This hypothesis is further supported by studies of ancestral nicotinic subunits[29], as discussed below.

Second, our data highlight the importance of concerted whole-body motions of the principal α subunit ECD, which appear to be essential to release potential conformational restraints at the α subunit interface between β1-β2 and M2-M3. Concerted whole-body motions of the α subunit ECD are likely facilitated by a conserved tripartite salt bridge involving a central αArg209 at the base of β10 and anionic residues in the β1-β2 loop (αGlu45), the β6-β7 loop (αAsp138), and the β8-β9 loop (αGlu175). Consistent with this hypothesis, mutations to any of these charged residues in the *Torpedo* nAChR lead to large changes in channel function, with mutations to the salt bridge between

αArg209 and αGlu45 having a particularly detrimental effect[6,9,15,20,21]. In contrast, mutations to analogous charged residues in non-α subunits, which do not undergo such agonist-induced whole body ECD motions, are remarkably well tolerated.

Third, unlike both α subunits, each non-α subunit adopts a locally active-like conformation at its intra-subunit ECD-TMD domain interface even prior to agonist binding. The subunit conformational asymmetry, however, is *not* maintained at the hydrophobic gate, with all five pore-lining M2 α-helices adopting a closed conformation that blocks cation flux. One interpretation is that the principal α subunits impose a global closed phenotype onto the entire TMD. This interpretation is supported by a study showing that a non-agonist binding muscle type ancestral β subunit ($\beta_{Anc}$) forms homo-pentamers that undergo spontaneous bursts of channel openings[29], while the incorporation of a muscle α subunit represses these spontaneous channel openings leading to a closed, yet agonist activatable channel[30]. The repression of spontaneous openings by the α subunit in the $\beta_{Anc}$ hetero-pentamers is consistent with our suggestion that the α subunit in the *Torpedo* nAChR imposes a closed conformation onto each non-α subunit TMD. We suggest that this internal local conformational asymmetry leads to local tension within the nAChR pentamer that is released upon agonist binding, and that this release of tension contributes to the relative energies of the closed and open states. The proposed release of local conformational tension may mimic the allosteric transitions that occur in hemoglobin, where oxygen binding relieves local tension in the porphyrin ring to shift hemoglobin from a low-affinity oxygen binding "tensed" to a high-affinity oxygen binding "relaxed" conformation[31].

Notably, the same release of conformational asymmetry is observed upon agonist binding to another heteromeric pLGIC (Fig. 6). In the $\alpha_1_2\beta_2_2\gamma_2$ GABA$_A$R, the αVal46 equivalent residue in the β1-β2 loop of the principal agonist binding β2 subunits, β2Val53, transitions from the pore distal to the pore proximal side of β2Pro273 of M2-M3 upon agonist binding, while the equivalent residues in the non-principal agonist binding subunits, α1His56/γIle68, are located on the pore proximal side of α1Pro278/γPro288 both prior to and after agonist binding[32,33]. As in the nAChR, this subunit conformational heterogeneity is not maintained in the TMD with the pore lining hydrophobic residues forming a relatively symmetric girdle that does not conduct anions. Agonist binding releases the conformational restraints so that the pentamer relaxes into a more symmetric open state. Mutagenesis data from the $\alpha_1_2\beta_2_2\gamma_2$ GABA$_A$R further support our gating model in that they highlight the functional importance of the αVal46 and αPro265 equivalent residues in principal agonist binding, but not in non-principal subunits. Specifically, β2V53A severely impairs channel gating, almost exclusively by slowing channel opening, while α1H55A has almost no effect[34,35]. Mutations to β2Pro273 also have much larger effects on channel function than similar mutations to α1P277A[36,37].

The homomeric α7 nAChR, 5-HT$_{3A}$R, α1 GlyR, ELIC, and GLIC also undergo similar structural rearrangements at each intra-subunit ECD-TMD domain interface[4,5,38-42] suggesting that a release of conformational restraints model applies to homomeric pLGICs, although such receptors do not exhibit local conformational asymmetry (Fig. 6). In the homomeric pLGICs, similar whole-body motions of the ECD may release each M2-M3 loop to relax outwards into an open conformation, with the resulting conformational transitions energetically governed by the energetics of wetting/de-wetting. Considerable data suggest that the αVal46 and αPro265 equivalent residues in these homomeric pLGICs play an important role in the channel function, although the effects of mutations to these residues are nuanced. For example, changing the αVal46 equivalent residue, Lys46, in the α7 nAChR to Cys or Glu[43] leads to a complete loss of function, as does the equivalent αP265G mutation in both the 5-HT$_{3A}$R and the α7 nAChR[44,45]. In contrast, mutations to the αVal46 equivalent in the α1

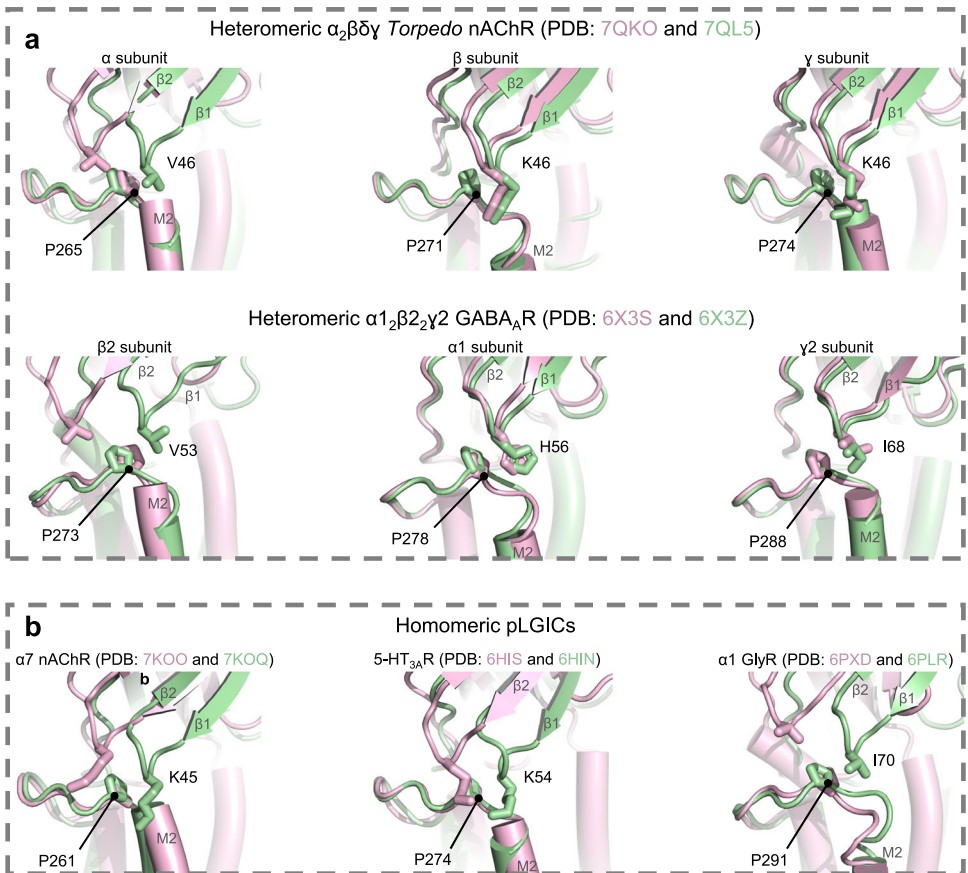

**Fig. 6 | Agonist-induced motions of the β1-β2 and M2-M3 loops at the ECD – TMD interface are a conserved feature of pLGICs.** Resting and agonist-bound conformations (pink and green, respectively) are shown in (**a**) for heteromeric pLGICs, the *Torpedo* nAChR (top)[12] and the human synaptic α1β2γ2 GABA_AR (bottom)[32], and in (**b**) for homomeric pLGICs, the human α7 nAChR[5], mouse 5-HT_3AR[38], and zebrafish α1 GlyR[4]). All structures are aligned by their M2-M3 loop to emphasize the relative motion of the β1-β2 loop.

GlyR have less of an effect[46]. Of particular interest, changing the αPro265 equivalent residue in (Pro246) GLIC to Gly leads to a large loss-of-function. Crystal structures grown under activating conditions (pH = 4.0) show that the P246G mutant adopts a conformation where the ECD remains in an active state but the pore collapses to a closed conformation with Lys32 now adopting a position on the pore distal side of Gly246[47]. This structure not only highlights the importance of Pro246 in GLIC gating but suggests that P246G may alter the structure of M2-M3 so that the TMD can easily transition between open and closed conformations, with crystallization capturing the TMD in a locally closed conformation. It remains to be determined whether subunit conformational asymmetry contributes to gating in homomeric pLGICs.

Finally, the concerted whole-body motions of the principal subunit ECD observed in the *Torpedo* nAChR are conserved in all pLGIC structures solved to date (see Movie S5 in ref. 10) As noted, charged residues in the *Torpedo* nAChR appear to facilitate these whole-body motions, with the salt bridge between αArg209 and αGlu45 playing a particularly important role. Mutations to the two analogous residues in the homomeric α1GlyR, ρ1GABA_AR, 5-HT_3AR, and α7 nAChR, have profound effects on channel function/expression, in many cases completely abolishing agonist-activation[43,48–51]. To our knowledge, the only heteromeric pLGICs where mutations to these residues have been characterized are the synaptic α1β2γ2 GABA_AR and the ganglionic α3β4 nAChR, where alanine/charge reversal mutations surprisingly have almost no effect[52–54]. The latter may indicate that the concerted movements of the ECD in these heteromeric pLGICs are facilitated by other interactions. In the

prokaryote, GLIC, this salt bridge is conserved (Arg192 and Asp31) and is critical to channel function, with the D31A mutant expressing robustly but failing to exhibit agonist-induced currents. In the prokaryote, ELIC, the αGlu45 equivalent residue is Thr28. Interestingly, the T28D mutation, which re-instates a salt bridge with Arg199 on the β10-M1 linker, leads to a 50-fold gain-of-function without any effect on agonist binding affinity[55].

In summary, we have elucidated a key feature underlying allosteric communication at the ECD – TMD interface in the heteromeric muscle nAChR. In the apo state, the β1-β2 loop extending down from the ECD of the principal agonist binding α subunit is positioned where it likely restrains the α subunit M2-M3 loop in a closed conformation while the same structures in non-α subunits already adopt a locally active conformation. Agonist binding leads to whole body movements of the principal subunit ECD that reposition the β1-β2 loop to release this local conformational heterogeneity and allow the α subunit M2-M3 loop to transition outward into a conformationally symmetric open state. Our data are consistent with early cryo-EM reconstructions of the *Torpedo* nAChR[56], reconcile an extensive body of mutagenesis data pertaining to the functional roles of residues at the ECD-TMD interface, and provide a framework for understanding allosteric communication at the ECD-TMD interface in other pLGICs.

## Methods
### Ethical statement
The experimental protocol involving the use of these animals has been approved by the Animal Care Committee of University of Ottawa (OHRI-2092). Ovulating female *Xenopus laevis* were purchased from

Nasco (Fort Atkinson, Wisconsin) and were housed in the animal care center of University of Ottawa, Faculty of Medicine.

## Molecular biology and electrophysiology

The following nAChR-pRBG4 cDNA clones were kindly provided by Steven Sine: Human α1, β1, δ, and ε, *Torpedo* α1, β1, γ, and δ. Each nAChR subunit cDNA was transferred into the pcDNA3.1 vector as an EcoRI fragment. Site-directed mutagenesis on these constructs were performed using QuikChange Lightning kits (Agilent), with primers used to make all mutants provided in a supplemental excel file. Following PCR, plasmids were transformed into *E. coli* Dh5α XL1-Blue/ Gold cells (Agilent) and these cells were then grown in super optimal broth for 1 h at 37 °C. The cells were then plated on agar plates containing ampicillin for 12–18 h at 37 °C. Individual colonies from these plates were then inoculated in 5–7 mL of ampicillin containing LB broth and grown over night in a shaking incubator at 37 °C. cDNA was then extracted and purified from the resulting cultures using a miniprep kit (Qiagen). The sequences of all constructs used were verified by Sanger sequencing (uLaval).

Once the sequences of the WT and mutant cDNA was confirmed, the plasmids were linearized using the restriction enzyme XhoI (New England Biolabs) just following the end of the coding sequence. Linearized DNA was then purified using a PCR purification kit (Qiagen). Mature cRNA was then synthesized using the mMESSAGE mMACHINE T7p in vitro transcription kit (Ambion). cRNA for the desired subunits were then mixed at a 2:1:1:1 ratio of α:β:δ:γ/ε at the desired concentration for oocyte injection. A total of 12.5 ng of cRNA was used for expressing the WT nAChR and up to 75 ng was injected for mutants that expressed poorly. All experiments working with RNA were performed on ice. All cDNA and cRNA was stored at −20 °C.

Defolliculated stage V-VI oocytes were injected with mRNA mixes for the mutant of interest and allowed to incubate for 2 to 5 days at 16 °C in ND96+ buffer (5 mM HEPES, 96 mM NaCl, 2 mM KCl, 1 mM MgCl$_2$, 1 mM CaCl$_2$, 2 mM pyruvate). Injected oocytes were placed in a RC-1Z oocyte chamber (Harvard Apparatus) containing HEPES buffer (96 mM NaCl, 2 mM KCl, 1.8 mM BaCl$_2$, 1 mM MgCl$_2$, 10 mM HEPES, pH 7.3). Whole-cell currents were recorded using a two-electrode voltage-clamp apparatus (OC-725C oocyte clamp; Harvard Apparatus) using the LabScribe v4 (iWorx) software. The whole-cell currents were recorded while the appropriate buffer flowed through the oocyte chamber at a rate of 15–20 mL/min. Currents through the plasma membrane in response to ACh concentration jumps (from 0 µM up to the indicated values) were measured with the transmembrane voltage clamped at −60 mV. Mutants that did not produce currents when injected with 75 ng of cRNA and incubated for 5 days were given the label non-responsive (NR) in the data tables. In certain cases, cell surface expression was measured (see below) and mutants that did not express were labelled non-expressed (NE) while those that did express but did not produce currents were labelled non-functional (NF).

Dose responses for each mutant were acquired from at least two different batches of oocytes. Peak current amplitudes from raw traces were extrapolated in OriginLab 2022. Each individual dose-response experiment was fit with a variable slope sigmoidal dose response using Prism 8.0.0 (GraphPad), and the individual EC$_{50}$ and Hill coefficients from each experiment averaged to give the values ± SD. A minimum of $n = 8$ experiments were collected wherever possible. Statistical significance was tested using a one-way ANOVA, followed by Dunnet's post hoc test. Degrees of freedom (DF) and F values for each ANOVA test are provided for each data table.

## Mutant cycle analysis

To determine the extent to which two or more residues contribute to channel function, the EC$_{50}$ values obtained from the mutagenesis experiments were cast as mutant cycles using Eq. 1:

$$\Omega = \frac{EC_{50}(WT) * EC_{50}(mut_{1,2})}{EC_{50}(mut_1) * EC_{50}(mut_2)} \qquad (1)$$

where WT is wildtype, mut$_1$ and mut$_2$ are the two single mutants, and mut$_{1,2}$ is the corresponding double mutant. The Ω represents how much the altered interaction contributes to channel function by reporting the deviation from independence. A pair that is completely independent will give a value of Ω = 1 with larger deviations from independence resulting in larger deviations from unity. To facilitate a comparison of these energetic couplings, we have also presented energy values in the tables using the Eq. 2:

$$\Delta\Delta G = RT\ln\Omega \qquad (2)$$

Where ΔΔG is the free energy of coupling, R is the universal gas constant, and T is 298 K. These values are not quantitative, however, as the EC$_{50}$ values from our measurement are composite values that are not representative of a single equilibrium constant in the activation mechanisms (i.e. channel gating). The values are instead used to illustrate the extent to which the residues are energetically coupled.

## Radioligand binding

Cell surface expression was measured where indicated on intact oocytes using [$^{125}$I]-α-bungarotoxin ([$^{125}$I]-BTX; PerkinElmer Life Sciences (Boston, MA)). Oocytes were injected with cRNA (50 ng) and allowed to express for 2 days. A minimum of four oocytes for each mutant were tested via two-electrode voltage clamp (TEVC) to determine activity levels, then incubated with occasional shaking, for 2 h at room temperature in MOR2 buffer (82 mM NaCl, 2.5 mM KCl, 1 mM Na$_2$HPO$_4$, 5 mM MgCl$_2$, 0.2 mM CaCl$_2$ and 5 mM HEPES, pH 7.4) with 2.5 nM [$^{125}$I]-BTX (143.8 Ci/mmol) and 1 mg/mL BSA. After incubation, the oocytes were washed 5 times with 2 mL of MOR2 buffer. [$^{125}$I]-BTX binding was quantified by γ-counting. Nonspecific binding was determined by the amount of toxin bound to mock-injected oocytes under the same conditions. Data presented in the tables was normalized to the number of counts for the WT nAChR. Mutants that expressed significantly more than mock injected oocytes were considered expressed. Statistical significance was tested using a one-way ANOVA, followed by Dunnet's post hoc test.

## Molecular dynamics simulations

Recently solved high resolution structures of the apo (PDB: 7QKO) and nicotine bound (PDB: 7QL5) states of the nAChR were used as input structures for simulations[12]. Systems were set up using CHARMM-GUI membrane builder[57]. Protein structures were embedded in a 140 × 140 Å pure POPC bilayer, with a box length of 190 Å, and solvated in TIP3P water and a NaCl concentration of 150 mM. Disulfide bonds were preserved as in the original structure. In-silico mutations were created using the CHARMM-GUI mutation tool. Simulations were performed using the GROMACS 2021 simulation engine[58,59] and the CHARMM36 forcefield[60,61] was applied. Each system was energy minimized for 5000 steps using the steepest descent algorithm or until a maximum force of 1000 kJmol$^{-1}$nm$^{-1}$ on any atom was reached. Equilibration was performed using the standard CHARMM-GUI protocol. Briefly, systems were equilibrated using the NPT ensemble for a total of 750 ps with a gradual lowering of protein and membrane position restraints. Temperature was held at 303.15 K using the Berendsen thermostat[62]. For the last four steps of equilibration, pressure was held at 1 bar using the Berendsen barostat[62]. Production runs were performed in the NPT ensemble with temperature set to 303.15 K using the Nose-Hoover thermostat[63,64] and pressure held at 1 bar using the Parinello-Rahman barostat[65]. Covalent bonds including hydrogen atoms were constrained using the LINCS algorithm[66]. The Verlet cut-off

scheme[67] was used throughout all steps with a force-switch modifier starting at 10 Å and a cut-off of 12 Å. The particle mesh Ewald (PME)[68,69] method was used for long-range electrostatics, and a cut-off of 12 Å was used for short-range electrostatics. The same simulation protocol was used for simulations of both states of the WT and mutant nAChR. Each system was simulated for 100 ns in triplicate. The MDAnalysis python package (version 2.0.0)[70,71] was used to create in-house scripts for simulation analysis. The distance between backbone CA atoms of the αV255 residues in was used as a measure of pore collapse. Plots were prepared using the python package matplotlib (version 3.8.2)[72].

### Model visualization and figure preparation
Visualization and image generation of molecular models was performed in PyMol version 2.5.4 (Schodinger) and ChimeraX version 1.5. All PDB entries used are provided in the Data Availability statement.

### Reporting summary
Further information on research design is available in the Nature Portfolio Reporting Summary linked to this article.

## Data availability
The data that support this study are available from the corresponding authors upon request. All analysed TEVC data is presented in the main text or supplemental information. Representative whole cell traces are shown in Fig. S2. Raw data used to create box and whisker plots in Figs. 2a, 3d, and 5c is provided in a Source Data file. *P* values for statistical tests in each data table are also provided in the Source Data file. Primers used to create every mutant characterized in the manuscript are provided in the Source Data file. Initial and final frames from all simulations are available Zenodo [https://doi.org/10.5281/zenodo.10582132]. The following PDB entries were used in the manuscript: 2BG9, 7QKO, 7QL5, 6X3S, 6X3Z, 7KOO, 7KOQ, 6HIS, 6HIN, 6PXD, 6PLR, 3TLS, 5HEG, and 5HEW. Source data are provided with this paper.

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

## Acknowledgements

This work was supported by grants from CIHR (175223) and NSERC (113312) to JEB and by an NSERC CGS-D scholarship to MJT.

## Author contributions

M.J.T. and J.E.B. designed the research project. M.J.T. and F.M.B. generated the mutants and acquired/processed the electro-physiology data. A.A. performed molecular dynamics simulations. M.J.T. and J.E.B. wrote the paper in consultation with H.N., with M.J.T. preparing all figures.

## Competing interests

The authors declare no competing interests.
