## [Peer Review File · Nature Communications]

A release of local subunit conformational heterogeneity underlies gating in a muscle nicotinic acetylcholine receptor.Reviewer #1 (Remarks to the Author):

The paper by Thompson, et al., addresses the mechanism by which binding of ACh to the nAChR is transduced into opening of its ion channel. Building on recently published structures of closed and putative open channel conformations of the prototypical Torpedo nAChR, the authors test inter-residue interactions revealed by these structures using mutagenesis combined with whole cell voltage clamp electrophysiology. The results show that inter-residue interactions within the alpha subunits are the primary mediators of the transductions process, including a network of electrostatic residues that remains intact during transduction, and an inter-loop interaction linked to transmembrane alpha helices that move during transduction. The overall results suggest the structural changes due to agonist binding release conformational restraints that hold the channel in the closed inactive conformation thus allowing the transmembrane alpha helices to dilate, opening the channel.

Many of the conceptual features of the binding transduction process reported here were already known, including the importance of capping of loop-C around bound agonist, the link between the capping motion to a network of conserved charged residues, communication of this network to the inter-loop interaction between the beta1-beta2 and M2-M3 loops, and subsequent outward motions of transmembrane alpha helices that line the pore. That these motions were crucial in the alpha subunits but not the non-alpha subunits was also known. In fact, the content in the first two paragraphs of the Results section (lines 79-112) and at least one more section (lines 248-257) are restatements of what is already known. Further, the whole cell electrophysiology, which cannot distinguish changes in agonist binding versus transductions steps, although qualitatively consistent with previous work, adds little to what is already known. A potentially intriguing physical interaction between the alpha subunits and the adjacent gamma and delta subunits is tested by mutagenesis and whole cell electrophysiology, but no evidence of inter-residue interactions contributing to transduction is detected, a negative result.

The structural change showing the Val46 residue from the beta1-beta2 loop jumping over the Pro265 residue in the M2-M3 loop, allowing release of conformational restraints, is intriguing. However, this could have been postulated based on the previously reported structures. That Pro265 and not Pro272 contacted Val46 has been clear since the first publication of high resolution nAChR structures. The electrophysiological analyses of the interaction between Val46 and Pro265 is equivocal, showing interaction values of omega of 1.33 for the pair Val46Ala plus Pro265Ala and 0.15 for the pair Val46Ala plus Pro265Gly. Hardly compelling evidence for functional importance; the functional interaction between these residues may have eluded detection by the authors use of whole cell rather than single channel measurements. In whole cell measurements the key readout is the EC50 for ACh, but the accuracy of this value depends on correctly determining the maximum response, which is underestimated due to rapid desensitization compounded by the inherently slow application of agonist in oocyte recording; whether this underestimate is the same for wild type and all mutants is unclear. Another perplexing issue is the author's claim that the jumping of Val46 over Pro265 in the Torpedo nAChR is also seen in the work on alpha7; I checked the published closed and open alpha7 structures but found that the residue equivalent to the Val46 (a Lys) does not jump over the residue equivalent to Pro265. Instead, the alpha7 structures show tighter engagement of the beta1-beta2 and M2-M3 loops in going from closed to open structures.

The extent of the advance by the proposed structural mechanism seems overstated. The major advance is to describe with residue level resolution the gating transition through comparison of high resolution closed and apparently open structures from published work. The proposed mechanism of release of conformational constraints is consistent with what was previously known and what is presented in the paper. The major value is the integration of current structural data with previous and newly presented mutagenesis/functional data.

Reviewer #2 (Remarks to the Author):

The manuscript by Thomson et al reports on an extensive mutagenesis analysis (~300 mutations) along with function characterizations by patch clamp electrophysiology to explore allosteric coupling at the human adult muscle nAChR. Despite extensive literature exists on this subject, this study is timely because the collected data are homogeneous (i.e., same subtype, composition, and expression system) and mutational effects can be interpreted considering the recently solved cryo-EM structures of the open and closed states of the receptors [Zarkadas 2022].

The results presented in this manuscript highlight the critical role of the ECD/TMD interfacial loops on coupling agonist-binding to channel opening, which is not novel per se. However, they suggest a mechanism where the agonist-induced conformational changes at the ECD are transmitted to the TMD via the "release" or removal of conformational restraints at the ECD/TMD interface. In combination with structural analyses of the non-alpha subunits, they suggest that the non-agonist-binding subunits are naturally primed or pre-activated even in the absence of agonist. Altogether, these results suggest a novel mechanism for channel activation in heteromeric channels, which is supported by structural/functional data at homologous pLGICs.

The mechanistic insight of this work is novel and interesting. However, some conclusions are not fully supported by data. Moreover, not enough credit is given to previous mechanistic studies particularly those based on simulations. Find below my series of points.

1. The authors refer to their mechanistic interpretation, i.e., the release of conformational restraints on the movement of the M2-M3 loop via agonist-induced movements of the b1-b2 loop at the ECD/TMD interface, as a novel proposal. I would argue that the same mechanism was proposed >10 years ago based on MD simulations of GluCl, i.e., Calimet et al 2013 (10.1073/pnas.131378511). Moreover, a very similar interpretation of available mutagenesis studies was given in that paper (see Supporting Information) to support that model of gating. In addition to that, the absence of strong or mechanical coupling between agonist binding and pore opening, which is referred to as "bevel-gear mechanism" by Thompson et al, was already discussed in Cecchini and Changeux 2015 (10.1016/j.neuropharm.2014.12.006), where such a mechanism is referred to as an "indirect coupling". None of these papers is mentioned.

2. On page 6 (and other places), the authors state that "agonist-induced motions of the b1-b2 loop (and thus V46) release the constraints holding the M2-M3 loop in a closed conformation, which allows it to relax outward to open the channel". This interpretation assumes that the closed pore configuration is higher energy or "tensed" and relaxes to the open channel when the b1-b2 loop moves away. The authors have no data on the thermodynamic stability or free energy of the open vs closed states of the pore and this interpretation remains purely speculative. In addition, the supposed stability of the open pore is in contradictions with free energy calculations done with GLIC by Zhu and Hummer [10.1073/pnas.1009313107]. To avoid confusion, I would avoid discussing the energetics whatsoever and replace "relax to an open conformation" with "form" or "adopt" throughout the manuscript.

3. Concerning pre-activation of the non-alpha subunits, it is stated on page 8 that "each non-alpha subunit adopts a locally active-like conformation at its intrasubunit ECD/TMD interface even prior to agonist binding". How local are these changes? An interesting hypothesis would be that the asymmetry of the preactivated non-alpha subunits extends to the entire ECD rather than being limited to the interfacial loops. In this respect, it would be good to know if pre-activation at non-alpha subunits involves capping of the C-loop, beta-contraction/extension, global tilting and twisting of the ECD subunits, etc.

4. The mutagenesis studies point to a critical role of the aArg209 in interaction with the nearby anionic residues aGlu45, aAsp138, aGlu175. Since charge reversal or annihilation at Arg209 results in no expression, it is unclear at this stage whether this interaction involves receptor folding or gating. In

addition, the location of these residues is reminiscent of the “switch of interactions” mechanism proposed in by Lev et al in GLIC [10.1073/pnas.1617567114]. In this paper, a switch of interaction between formally charged residues was proposed to mediate GLIC function by promoting tertiary changes at the lower beta-sandwiches, referred to as beta-contraction/expansion. Are similar changes observed in the muscle nAChR structures? Is there a switch of interactions involving Arg209? If so, this model should be discussed and the work of Lev et al mentioned.

5. The data in Fig.4C are puzzling. First, there is no panel C in the caption of Fig.4. Second, the large loss-of-function shown for aE45R in Fig.4C is inconsistent with the data reported in Table I (i.e., 15.6 gain-of-function). Third, the data shown in Fig.4C do not appear in Table I (i.e., aD138A, aE175A, etc.). Please correct/amend.

Minor points

- Page 6, line 254 “Mutations to the coordinated anionic residues were better tolerated than those of α Arg209, presumably because they can partially compensate for the loss of each other, but still lead to relatively large changes in EC50 values.”: Where does one find the data?
- Page 7, line 286: usually
- Page 8, line 351: “equivalent residues in principal agonist binding, but not principal agonist-binding subunits”. Unclear, consider revising.
- Page 9, line 369: include Ref to locally-closed structures of GLIC.

Summary of Reviewers' comments and response

We thank both reviewers for their extensive and insightful comments which have helped improved the manuscript substantially. We have carefully considered all the critiques of our manuscript and have addressed the noted concerns in the revised text, as summarized in the point-by-point response to the reviewers' comments below. We have also revised figures bar plots to include violin plots and have ensured that appropriate citations to all data tables, etc., are included in the text and figure legends. We have included new movies that more clearly illustrate the structural motions that are central to the discussion, have included raw electrophysiological traces for key mutants (supplemental information), and have performed extensive analyses of the conformational changes that take place in the extracellular domain of the *Torpedo* nAChR upon agonist-binding.

We hope that the revised text will satisfy the reviewers' concerns. We look forward to the reviewers' responses and are happy to consider additional changes that will bring the manuscript to publication.

Although both reviewers acknowledged strengths in the manuscript, Reviewer #1 expressed a central concern that the building blocks of the work are not new. We have revised the text to better highlight the known building blocks and how we have assembled these in the context of new structural and functional observations to solve an important question in pentameric ligand-gated ion channel (pLGIC) biology. Reviewer #2 found the study 'timely' and 'coherent'. They noted that the "mechanistic insight of this work is novel" yet underlined a global oversight in the lack of discussion of previous studies performed using molecular dynamic simulations. In response, we have explicitly discussed our work in context of these previous findings. We have also examined in detail the changes in structure of the ECDs upon agonist binding, as presented below in the rebuttal.

Point-by-point response to Reviewers' comment

Quotes from the Reviewer's comments appear in dark green. Responses are in black, with quotes from the text highlighted in Italics. In sections where we have added important new text to previously established text, we highlight the new additions with a yellow background.

Reviewer #1:

The paper by Thompson, et al., addresses the mechanism by which binding of ACh to the nAChR is transduced into opening of its ion channel. Building on recently published structures of closed and putative open channel conformations of the prototypical Torpedo nAChR, the authors test inter-residue interactions revealed by these structures using mutagenesis combined with whole cell voltage clamp electrophysiology. The results show that inter-residue interactions within the alpha subunits are the primary mediators of the transductions process, including a network of electrostatic residues that remains intact during transduction, and an inter-loop interaction linked to transmembrane alpha helices that move during transduction. The overall results suggest the structural changes due to agonist binding release conformational restraints that hold the channel in the closed inactive conformation thus allowing the transmembrane alpha helices to dilate, opening the channel.

Many of the conceptual features of the binding transduction process reported here were already known, including the importance of capping of loop-C around bound agonist, the link between the capping

motion to a network of conserved charged residues, communication of this network to the inter-loop interaction between the beta1-beta2 and M2-M3 loops, and subsequent outward motions of transmembrane alpha helices that line the pore.

We agree with the reviewer that many of the conceptual features underlying allosteric communication between the agonist site and channel gate are well established and apologize for the lack of clarity in our Introduction as to the precise goals and rationale for our study. We have revised the Abstract and Introduction to clarify both the established features and those that have remained enigmatic. Specifically, we highlight the gaps in our understanding of how structural changes in the extracellular domain (ECD) are directly communicated across the ECD-TMD interface to the transmembrane domain (TMD) to open the channel gate, which is the central focus of this work (major changes in yellow):

Abstract:

“Synaptic receptors respond to neurotransmitters by opening an ion channel across the post-synaptic membrane to elicit a cellular response. Here we use recent structures of the Torpedo acetylcholine receptor and functional measurements to delineate a key feature underlying allosteric communication between the agonist-binding extracellular and channel-gating transmembrane domains. Extensive mutagenesis singles out the established energetically coupled role for the principal α subunit β 1- β 2 and M2-M3 loops, with agonist binding re-positioning a key β 1- β 2 glutamate/valine so that a conserved M2-M3 proline translates outward to open the channel gate. Notably, the analogous structures in non- α subunits adopt a locally active-like conformation in the apo state even though the hydrophobic gate in all five pore-lining M2 α -helices is closed. Agonist binding releases internal conformational heterogeneity transitioning all five subunits into a conformationally symmetric open state. A release of conformational heterogeneity provides a new framework for understanding allosteric communication across the inter-domain interface”.

Introduction (lines 44-75):

“While increasing structural data highlight the tertiary/quaternary motions that occur upon agonist binding to pLGICs¹⁻⁵, we have remained intrigued by a model originally based on the pioneering 4 Å resolution cryo-electron microscopy reconstruction of the muscle-type Torpedo nAChR (PDB code 2BG9)⁶. This structure revealed a “molecular continuum” in the principal α_v/α_s subunits leading from each agonist binding site to a critical salt bridge between α Arg209 at the base of β 10 and α Glu45 at the tip of the β 1- β 2 loop, with α Glu45 and the adjacent α Val46 straddling α Pro272 on the M2-M3 loop (Fig. 1b)⁶. This continuum suggested a bevel-gear type mechanism whereby the closing of loop C around the agonist translates into a conformational change in the bevel, the β 1- β 2 loop and the covalent link between β 10 and M1, that couples with movements of the gear, the M2-M3 loop, to open the pore-lining M2 α -helix.

Although many of the conceptual features central to this model are now well established, how the bevel couples directly with the gear to open the channel gate remains enigmatic. On the one hand, compelling data show that the above noted and other residues at the interface between β 1- β 2 and M2-M3 interface in the muscle α subunit are not only essential for, but couple energetically with each other during channel gating⁷⁻⁹. On the other hand, higher resolution structures now position α Pro272, and other implicated side chains, four residues further along the M2-M3 loop, where they are distant from β 1- β 2. The structures also show that the tip of β 1- β 2 not only does not engage tightly with, but moves roughly orthogonal to M2-M3 upon agonist binding (Fig. 1c)^{4,5,10-14}. The structural data are not compatible with

the bevel-gear type gating mechanism that is foundational to many models of pLGIC gating. In fact, there is a perplexing disconnect between functional studies, which identify critical residues and/or interacting partners at the ECD-TMD domain interface that regulate function in pLGICs, and recent structural data, which show that these same critical and/or interacting partners move away from each other upon agonist binding.

Further complicating our understanding of allosteric communication across the ECD-TMD interface, the extended F loops from the complementary γ/δ subunits in the Torpedo nAChR rock in upon agonist binding, which causes the membrane juxtaposed $\beta 8$ - $\beta 9$ loops/ $\beta 10$ -M1 linkers to pivot outward. Significantly, the $\beta 8$ - $\beta 9$ loops/ $\beta 10$ -M1 linkers from both γ and δ tightly sandwich the M2-M3 loops from the adjacent α_γ and α_δ subunits, with these structures all moving in concert to open the L9' gate (Movie 1). Both the tight interactions and the concerted motions at this inter-subunit ECD-TMD domain interface raise the possibility that it is the γ/δ $\beta 8$ - $\beta 9$ loops/ $\beta 10$ -M1 linkers, not the $\alpha_\gamma/\alpha_\delta$ $\beta 1$ - $\beta 2$ loops, that primarily couple the movements of the ECD to those of M2-M3 in the TMD during channel gating. In addition, instead of specific side chain interactions, it has been suggested that either the overall charging pattern or nonspecific bumping of closely apposed domains govern allosteric communication at the ECD-TMD interface^{15,16}."

That these motions were crucial in the alpha subunits but not the non-alpha subunits was also known.

We agree with the reviewer in that Lee and Sine (Nature, 2005 438, 243-247) initially showed that residues at the ECD-TMD interface of the α subunit are critical to channel function and exhibit strong energetic couplings, while mutations to analogous residues at the intra-subunit ECD-TMD interface in other subunits have little effect. As noted above and in the text of the manuscript, however, the mechanistic interpretation of these functional observations has been clouded by higher resolution structures, which both alter the relative positions of implicated residues and show that residues on either side of the ECD-TMD interface undergo roughly orthogonal movements to each other upon agonist binding. Our integrated structural and functional approach led us to re-focus our attention on the movements that occur at the ECD-TMD interface in principal α versus non- α subunits despite the available structural data suggesting that motions at the inter-subunit interface may be key to gating. In fact, the resulting structural observations, supported by additional functional studies, provide a mechanistic framework for understanding the functional asymmetry noted by Lee and Sine. To further highlight the seminal work of Lee and Sine, we added the following text to line 248-250.

"The local subunit conformational heterogeneity rationalizes previous functional data, which highlighted an important functional role for residues at the intra-subunit ECD-TMD interface of α , but not non- α subunits⁷".

Note that we also extended the functional data to the tripartite salt bridge centered around the α Arg209-equivalent residue in each subunit and show that this salt bridge plays a critical functional role in α but not non- α subunits, as discussed below.

In fact, the content in the first two paragraphs of the Results section (lines 79-112) and at least one more section (lines 248-257) are restatements of what is already known.

Although this comment is true, the data presented in the first two paragraphs of the results is only a minor part of the study, yet is foundational to our findings for the following reasons:

- 1) As our functional studies were designed to help interpret the *Torpedo* structural data, we felt it was essential to avoid ambiguity (see point 2 below) and thus assess the functional impact of *all* mutations directly in the *Torpedo* nAChR, even if similar mutations had already been characterized in human adult or murine versions.
- 2) As noted, a central rationale for this study was the disconnect between seminal functional studies, which identify critical residues and/or interacting partners at the α subunit β 1- β 2 – M2-M3 interface, and structural data, which show that these same critical and/or interacting partners move away from each other upon agonist binding. As the cited seminal functional data is from the human adult muscle nAChR (Lee and Sine, 2005) while the structural data is from the *Torpedo* nAChR, we could not a priori rule out the possibility that the apparent disconnect between structure and function reflects subtle mechanistic differences in gating between the two nAChR subtypes. By recording data from key mutants using both human adult muscle and *Torpedo* nAChRs expressed in oocytes, we show that these residues perform similar functional roles in both receptor subtypes and thus that the noted disconnect is instead due to a lack of understanding of the mechanistic underpinnings regarding the roles of these residues in channel function.
- 3) As pointed out by the reviewer and discussed in the text, changes in agonist EC₅₀ values cannot be interpreted exclusively in terms of effects on channel gating. Our study, however, has the benefit that the direct effects of some mutants on channel gating have already been assessed using single channel measurements, albeit in the human adult nAChR. By showing that identical mutations have similar effects on the function of both the human adult and the *Torpedo* nAChRs expressed in oocytes, we have a stronger rationale for incorporating the single channel measurements from human adult nAChRs into our mechanistic interpretations of the *Torpedo* structures.
- 4) Finally, a critical feature of this manuscript is the comparison between the functional consequences of mutations at the intra-subunit interface (i.e., at the interface between β 1- β 2 and M2-M3 in the α subunit as described in the first two paragraphs of the results) to those of mutations at the inter-subunit interface (i.e., at the interface between M2-M3 from the α subunit and β 8- β 9/ β 10-M1 from the γ / δ subunits). This comparison is essential because, as noted above and in the text, the movements of the β 8- β 9/ β 10-M1 from γ / δ at the inter-subunit interface correlate with the movements of M2-M3 from α that open the channel gate, while those of β 1- β 2 from α at the intra-subunit interface do not. Our extensive functional data, however, show that the residues at the inter-subunit interface are not critical for function and are less impactful on function than those at the intra-subunit interface. The latter comparison is the most important functional observation of this study and is foundational to the novel mechanistic understanding that we obtain regarding the nature of allosteric communication at the ECD-TMD interface. An unequivocal comparison of the functional effects of mutations at both the intra- and inter-subunit interfaces required that we perform equivalent mutations at both interfaces using the *Torpedo* nAChR for all functional observations.

To underscore these important points, we have added the following text to the manuscript. In the Introduction, we added the following statement to emphasize the need to mitigate complications arising from comparing data using different techniques with different nAChRs expressed in different cell types (lines 78-84):

“To explore the large number of potential interacting side chains, mitigate the complications arising from integrating mutagenesis data obtained using whole cell versus single channel recordings of muscle nAChRs from different species and with different subunit compositions (fetal versus adult) expressed in different heterologous systems, and to ensure that our functional data is directly relatable to the Torpedo structures, we used a screening mutagenesis approach with each mutation functionally characterized in the Torpedo nAChR expressed in Xenopus oocytes using two electrode voltage clamp electrophysiology”.

In the first paragraph of the results, we also state the following (lines 95-105):

“The compelling functional data suggesting a bevel-gear type gating mechanism involving α Arg209, α Glu45, and α Val46 from the ECD (referred to as the ECD triad) interacting with α Pro272 of M2-M3 was obtained from single channel recordings of the human adult muscle nAChR⁷, while the structural data revealing roughly orthogonal agonist-induced motions of β 1- β 2 and M2-M3 were obtained using the Torpedo nAChR¹⁰. To definitively rule out the possibility that these apparent discrepancies arise from subtle mechanistic differences between the human adult and Torpedo forms, we compared the functional consequences of key mutations in both receptors expressed in frog oocytes (Table 1). We observed essentially identical functional consequences when each of these residues was mutated suggesting that each residue contributes similarly to channel function in both the adult muscle and Torpedo nAChRs (Fig. 1d & S1c and Table 1)”.

With regards to the content of lines 248-257, we performed the extensive mutagenesis screen of residues contributing to the tripartite salt bridge because we hypothesized that pivoting motions of the α subunit ECD are critical to a release of conformational restraints, and that the tripartite salt bridge is critical to ensure that the ECD moves as a whole body. In contrast, our structures show that non- α subunits do not undergo this pivoting motion because they are already in a local active-like state at the ECD-TMD interface – thus suggesting that the tripartite salt bridge is not important. By mutating each of the residues involved in the tripartite salt bridge in each of the subunits, we provide evidence in support of our hypothesis. Note, however, that the figure in our manuscript describing these studies (Figure 4) focuses on the pivoting motions of the tripartite salt bridge in α versus non- α subunits, with the functional data representing only a minor part of the discussion. In fact, the table of the functional data is only included in the Supplementary information (Table S6).

We agree that it is well established that the salt bridge between α Arg209 and α Glu45 is critical to channel function, as originally shown for the human adult muscle nAChR by Lee and Sine (Nature, 2005 438, 243-247). Many, but not all, of the charged residues that contribute to the tripartite salt bridge in the various subunits have also been characterized in the following publications: Chakrapani et al. (J. Gen. Physiol., 2004 123, 341-356), Xiu et al. (JBC, 2005 280, 41655-41666), Bruhova and Auerbach (JBC, 2010 285, 38898-38904), Jha et al. (J. Physiol., 2012 590, 119-129), Mukhtasimova and Sine (Biophys. J., 2013 104, 355-367), Shen et al. (JBC, 2016 291, 3291-3301), and Shen et al. (JCI Insights, 2018 3, e97826). Note that the results from these studies are diverse and are not directly comparable as the studies were performed on different subtypes (human adult, murine fetal, and murine adult) using different

techniques (TEVC vs single channel) and even different activating agonists (ACh vs choline). Here we performed an extensive characterization of mutations to each of the four residues contributing to the tripartite salt bridge in each of the four distinct subunits of the *Torpedo* nAChR. In addition, we mutated these residues to a variety of side chains with different properties. While we acknowledge that our data, for the most part, is similar to that reported by others, we felt and still feel that including the mutagenesis screen in the supplemental information is appropriate for the sake of completeness (all four residues in each subunit) and because our functional data was obtained using the *Torpedo*, as opposed to human adult or murine adult/fetal, nAChR and is thus directly relevant to the *Torpedo* structures.

We have revised the text of the manuscript to state (line 271-275):

*“These concerted movements are likely facilitated by interactions within the ECD β sandwich including a conserved tripartite salt bridge between the α Arg209 and the anionic residues α Glu45 (β 1- β 2), α Asp138 (β 6- β 7), and α Glu175 (β 8- β 9). As noted, previous studies have highlighted a critical functional role for the salt bridge between α Arg209 and the anionic residues α Glu45 (β 1- β 2)^{7,15,21–26}. In the *Torpedo* nAChR, ...”*

We also state (lines 283-286):

“Given the lack of whole-body pivoting motions of each non- α subunit ECD, we hypothesized that the tripartite salt bridge in β , γ , and δ would not play a critical role in channel function. Consistent with this hypothesis and in agreement with previously published data^{7,15}, mutations to the analogous charged residues in these non- α subunits...”

Further, the whole cell electrophysiology, which cannot distinguish changes in agonist binding versus transductions steps, although qualitatively consistent with previous work, adds little to what is already known.

The reviewer is correct that EC₅₀ values reflect both binding and gating, as stated in the manuscript. The mutations studied here are distant from the agonist site and thus do not contribute directly to binding. In fact, this has already been documented in Supplemental Table S1 in Lee and Sine (Nature, 2005 438, 243-247) where mutations to residues at the ECD – TMD interface have little effect on agonist binding affinity. We discuss the correlation between single channel and whole cell measurements in the second paragraph of the Results (lines 116-120):

*“Note that although the measured variations in the EC₅₀ values relative to wild type for each mutation are similar in the *Torpedo* and adult muscle nAChRs, they differ substantially from the relative changes in the di-liganded gating equilibrium constants (θ) calculated for each mutation from kinetic fitting of single channel measurements (Fig. S1)⁷. Such differences, however, are expected because EC₅₀ values, which reflect both agonist binding (K_D) and channel gating (θ), are mathematically weighted towards the K_D ¹⁸”.*

Furthermore, we now present representative whole cell traces for key mutants analyzed in our study in Figure S3. These traces show that none of the mutants alters substantially the rates of desensitization.

Our data, however, must be evaluated in the context that we needed to screen an extremely large number of mutations at the ECD-TMD interface to gain the presented mechanistic insight. The advantageous ability to screen a large number of mutants using whole cell measurements in oocytes

has been noted by others (see, for example, Xiu et al (JBC 2005 280, 41655-41666)). In addition, by performing functional studies of both human and *Torpedo* nAChRs (first two paragraphs of the Results), we can draw similarities between the two receptor types, which gives us a better rationale for integrating findings from single channel measurements performed on the human adult nAChR directly into the interpretations of our *Torpedo* structural data. As pointed out by the reviewer, a unique feature of our paper is the integrative structural and functional approach. This approach led us to re-focus on the intra-subunit interface despite the structural data suggesting that this interface is not critical for channel function – at least in the context of the typical bevel-gear type mechanism usually invoked to understand allosteric communication at the ECD-TMD interface. Our integrated structural and functional approach led to the release of conformational restraints model presented in this work.

A potentially intriguing physical interaction between the alpha subunits and the adjacent gamma and delta subunits is tested by mutagenesis and whole cell electrophysiology, but no evidence of inter-residue interactions contributing to transduction is detected, a negative result.

We agree that this is a negative result but reiterate that the mutagenesis data must be evaluated in the broader context of the work. As noted above, a key element of our study is the comparison of the functional consequences of mutations at the intra-subunit interface to those at the inter-subunit interface. This comparison is essential because the movements at the inter-subunit interface correlate with opening the pore thus suggesting an important mechanistic role, while those at the intra-subunit interface do not. In contrast to the structural data, a key observation of our study is that the residues at the inter-subunit interface are not critical for and are less impactful on function than those at the intra-subunit interface. This important contradiction ultimately led us to focus our attention on the conformational transitions at the intra-subunit interface leading to the novel mechanistic understanding presented in this work. To emphasize this fact, we state in the Discussion (lines 302-315):

“A key finding of our mutagenesis data is that the intra- α subunit interactions between the β 1- β 2 and M2-M3 loops are functionally more important and exhibit stronger energetic couplings than the inter-subunit interactions between the β 8- β 9 loops/ β 10-M1 linkers from γ/δ and M2-M3 loops from $\alpha\gamma/\alpha\delta$. These findings are surprising given that the two α subunit β 1- β 2 loops do not engage tightly with and move roughly orthogonally to the motions of the α subunit M2-M3 loops, whereas the β 8- β 9 loops/ β 10-M1 linkers from γ/δ interact extensively and move in concert with the adjacent $\alpha\gamma/\alpha\delta$ M2-M3 loops. The contradictory structural and functional observations are not easily reconciled in the context of the bevel-gear type gating mechanism that is typically usually used to frame agonist-induced channel gating. In such a model, agonist binding is assumed to lead to movements of the ECD that result in the formation of new or enhanced interactions across the ECD-TMD domain that energetically stabilize the open pore. Although such interactions may occur, the inability to identify consensus interactions across the ECD-TMD interface that clearly define the open state, despite the extensive mutagenesis studies performed here and elsewhere^{15,16}, suggests that such open state defining interactions do not exist. Instead, we propose that...”

The structural change showing the Val46 residue from the beta1-beta2 loop jumping over the Pro265 residue in the M2-M3 loop, allowing release of conformational restraints, is intriguing. However, this could have been postulated based on the previously reported structures.

The Reviewer is correct that the 'jumping over' motion in principal subunits could have been postulated based on previously reported structures. In fact, as pointed out in Figure 5, this same jumping over motion is observed in all principal agonist binding subunits in pLGIC structures that have been solved in both apo and agonist-bound forms. Despite this, none of the papers reporting these structures has highlighted this motion nor has any publication noted either the absence of this motion in non principal subunits or that these subunits already adopt a locally “active” conformation in the apo state. It was our extensive screening mutagenesis that refocused our attention on the intra-subunit interface and that led to these structural findings. The fact that we detected these motions where others have not speaks to the importance of the functional measurements that we have obtained *even in the context of the stated limitations with the interpretation of EC₅₀ values.*

Note also that these motions are difficult to observe in the context of the complex quaternary/tertiary motions that occur upon agonist binding. Our extensive functional screening led us to focus on the intra-subunit ECD-TMD interface. It was only after careful alignments of the local structures that these relative motions became apparent.

That Pro265 and not Pro272 contacted Val46 has been clear since the first publication of high resolution nAChR structures.

It is well established that Val46 contacts Pro265, not Pro272. We noted this in the Introduction (lines 57-58):

“On the other hand, higher resolution structures now position α Pro272 and other implicated side chains four residues further along the M2-M3 loop, where it is distant from β 1- β 2”

Lee and Sine (Lee and Sine, Nature 2005 438, 243-247), however, show clearly that Val46 is critical to channel function and that it couples energetically with Pro272. As noted above, the new higher resolution structures clouded the mechanistic interpretation of this observation by Lee and Sine. Our study sheds light on the mechanistic underpinnings of these functional observations.

The electrophysiological analyses of the interaction between Val46 and Pro265 is equivocal, showing interaction values of omega of 1.33 for the pair Val46Ala plus Pro265Ala and 0.15 for the pair Val46Ala plus Pro265Gly. Hardly compelling evidence for functional importance; the functional interaction between these residues may have eluded detection by the authors use of whole cell rather than single channel measurements. In whole cell measurements the key readout is the EC₅₀ for ACh, but the accuracy of this value depends on correctly determining the maximum response, which is underestimated due to rapid desensitization compounded by the inherently slow application of agonist in oocyte recording; whether this underestimate is the same for wild type and all mutants is unclear.

There is an extensive electrophysiological literature characterizing mutants at the ECD – TMD interface, but it has remained difficult to rationalize all the data in terms of a cohesive model because of the variabilities in how different sets of data are acquired in different labs – i.e., different labs use single channel vs whole cell, different heterologous expression systems, different agonists, etc.

We chose to use whole cell recordings with nAChR mutants expressed in oocytes because this is the only approach that allowed us to screen the large number of mutations required to investigate the roles of a vast number of residues at the ECD – TMD interface in channel function using an internally consistent approach. We frequently relate our functional data to published data obtained using single channel measurements where the impact of a mutant on channel opening and closing rates can be directly assessed.

While our interaction omega values are closer to unity (i.e., less energetic coupling) than those derived from single-channel recordings, this is expected because of the smaller fold changes obtained from EC₅₀ values vs θ (gating equilibrium constant) values, as discussed in the text of the manuscript. For example we measure an omega value of 0.46 for the pair α Val46Ala and α Pro272Gly, which expectedly suggests a lower degree of energetic coupling than the omega value for the same pair of 0.06 obtained from single channel measurements. The latter omega value, however, was deemed “substantial” (Lee and Sine, Nature 2005 438, 243-247). Given our measured omega value between the pair α Val46Ala and α Pro265Gly is further from unity (0.15) than our measured omega value between the pair α Val46Ala and α Pro272Gly (0.46), it is likely that the energetic coupling between α Val46Ala and α Pro265Gly is larger than the energetic coupling between α Val46Ala and α Pro272Gly and is thus also substantial. Ultimately, the essence of our study is that the mutagenesis data led to structural observations that rationalize the huge functional importance of residue interactions at the intra- α subunit β 1- β 1 - M2-M3 interface that have been established using single channel measurements, particularly in both the Sine and Auerbach labs.

Another perplexing issue is the author’s claim that the jumping of Val46 over Pro265 in the *Torpedo* nAChR is also seen in the work on alpha7; I checked the published closed and open alpha7 structures but found that the residue equivalent to the Val46 (a Lys) does not jump over the residue equivalent to Pro265. Instead, the alpha7 structures show tighter engagement of the beta1-beta2 and M2-M3 loops in going from closed to open structures.

We believe this is primarily a visualization issue. As pointed out, visualization of these local motions in the context of the complex tertiary/quaternary motions that occur upon agonist binding to a pLGIC requires careful/specific alignment of the individual subunits. As shown in Fig. 5B of the manuscript, the motions of the α Val46 equivalent in the α 7 nAChR, Lys45, relative to the α Pro265 equivalent residue, Pro261, upon agonist binding are essentially superimposable on the motions of α Val46 relative to α Pro265 in the *Torpedo* nAChR.

To better illustrate the consistency of the motions upon agonist binding in all pLGICs, we have generated movies from two different orientations that show the morphed movements of $\beta 1$ - $\beta 1$ versus M2-M3 between apo and agonist-bound states in a variety of pLGICs, including the $\alpha 7$ nAChR (Movies 4 and 5). The movies clearly show that the motions of the α subunit $\beta 1$ - $\beta 1$ relative to M2-M3 are essentially identical in the $\alpha 7$ and *Torpedo* muscle-type nAChRs. Below are side views of the starting and end points of the transition in $\alpha 7$ superimposed on those of the *Torpedo* (please see movie 5 for a top down view):

The extent of the advance by the proposed structural mechanism seems overstated. The major advance is to describe with residue level resolution the gating transition through comparison of high resolution closed and apparently open structures from published work. The proposed mechanism of release of conformational constraints is consistent with what was previously known and what is presented in the paper. The major value is the integration of current structural data with previous and newly presented mutagenesis/functional data.

We respectfully disagree on a global overstatement of novelty. To our knowledge, previous structural and mutagenesis studies have not suggested that a release of conformational restraints underlies allosteric communication at the ECD-TMD interface. Our study builds on previous findings and uses an integrative structural and functional approach to develop a new model of allosteric communication. We would like to point out the numerous novel aspects of the study:

- 1) We present extensive electrophysiological data showing the functional importance and energetic couplings between residues at the $\beta 1$ - $\beta 1$ - M2-M3 interface of α subunits in both *Torpedo* and human adult nAChRs expressed in oocytes,
- 2) We present extensive functional data showing that residues at the interface between $\beta 1$ - $\beta 2$ and M2-M3 loops of the α subunit are functionally more important and exhibit stronger energetic couplings than those at the interface between the $\beta 8$ - $\beta 9$ loops/ $\beta 10$ -M1 linkers from γ/δ and M2-M3 loops from $\alpha_\gamma/\alpha_\delta$, even though the latter structures move in concert upon agonist binding,

3) Our detailed functional data led us to detect and thus highlight the relative movements of $\beta 1$ - $\beta 1$ versus M2-M3 in principal subunits, with functional data further highlighting the functional importance of these movements,

4) Our functional studies led us to detect the asymmetry in the movements of $\beta 1$ - $\beta 1$ versus M2-M3 in principal versus complementary subunits,

5) Our functional studies guided a detailed analyses of published structures which led us to identify the conservation of these movements across the pLGIC superfamily

6) Our functional studies guided a detailed analyses of published structures and led us to identify that there is asymmetry in the intra-subunit conformations of complementary subunits in the apo state whereby they adopt a conformation that is locally active at the $\beta 1$ - $\beta 1$ - M2-M3 interface but locally closed in the TMD.

7) Our functional studies guided a detailed analyses of structures leading to the realization that there is a release of this conformational asymmetry upon agonist binding and the opening of the L9' gate.

We believe these structural observations have never been previously discussed together in the literature, nor individually for points 3-7.

Nevertheless, we have revised the text to tone down the vocabulary related to novelty and to focus on the key findings. Indeed, the reviewer has effectively conveyed that the specialized audience that we target, beyond the general reader interested in allosteric transitions, will evaluate the novelty of these findings in the context of the known literature without any bravado statements from us. For example, we have revised the Abstract from:

"Here we use recent structures of the Torpedo acetylcholine receptor and screening mutagenesis to delineate the mechanisms underlying the coupling of agonist binding to channel gating".

to more appropriately state:

"Here we use recent structures of the Torpedo acetylcholine receptor and functional measurements to delineate a key feature underlying allosteric communication across the interface between extracellular and transmembrane domains".

We also changed the introduction from:

"A release of conformational asymmetry model not only defines gating in the muscle nAChR but reconciles..."

to state:

"The release of conformational asymmetry is not only a key feature underlying allosteric communication at the ECD-TMD interface in the muscle nAChR but reconciles..."

Other changes are evident in the tracked-change version of the manuscript.

Reviewer #2 (Remarks to the Author):

The manuscript by Thomson et al reports on an extensive mutagenesis analysis (~300 mutations) along with function characterizations by patch clamp electrophysiology to explore allosteric coupling at the human adult muscle nAChR. Despite extensive literature exists on this subject, this study is timely because the collected data are homogeneous (i.e., same subtype, composition, and expression system) and mutational effects can be interpreted considering the recently solved cryo-EM structures of the open and closed states of the receptors [Zarkadas 2022].

The results presented in this manuscript highlight the critical role of the ECD/TMD interfacial loops on coupling agonist-binding to channel opening, which is not novel per se. However, they suggest a mechanism where the agonist-induced conformational changes at the ECD are transmitted to the TMD via the “release” or removal of conformational restraints at the ECD/TMD interface. In combination with structural analyses of the non-alpha subunits, they suggest that the non-agonist-binding subunits are naturally primed or pre-activated even in the absence of agonist. Altogether, these results suggest a novel mechanism for channel activation in heteromeric channels, which is supported by structural/functional data at homologous pLGICs.

The mechanistic insight of this work is novel and interesting. However, some conclusions are not fully supported by data. Moreover, not enough credit is given to previous mechanistic studies particularly those based on simulations. Find below my series of points.

1. The authors refer to their mechanistic interpretation, i.e., the release of conformational restraints on the movement of the M2-M3 loop via agonist-induced movements of the b1-b2 loop at the ECD/TMD interface, as a novel proposal. I would argue that the same mechanism was proposed >10 years ago based on MD simulations of GluCl, i.e., Calimet et al 2013 (10.1073/pnas.131378511). Moreover, a very similar interpretation of available mutagenesis studies was given in that paper (see Supporting Information) to support that model of gating. In addition to that, the absence of strong or mechanical coupling between agonist binding and pore opening, which is referred to as “bevel-gear mechanism” by Thompson et al, was already discussed in Cecchini and Changeux 2015 (10.1016/j.neuropharm.2014.12.006), where such a mechanism is referred to as an “indirect coupling”. None of these papers is mentioned.

The lack of a citation of these two papers was an oversight on our part. We have revised the Introduction to state:

“While increasing structural data highlight the tertiary/quaternary motions that occur upon agonist binding to pLGICs¹⁻⁵, we have remained intrigued by a model...”

We have cited the review by Cecchini and Changeux (Neuropharmacology, 2015 96, 137-149) in line 45. We have also added the following sentence to the Results section (lines 267 & 268):

“Similar motions of $\beta 1$ - $\beta 2$ relative to M2-M3 have also been suggested previously to underlie gating in the glutamate activated chloride channel, GluCl²⁰”

and have referenced the work of Calimet et al. (PNAS, 2013 110, E3987-E3996).

It is important to note that Calimet et al. proposed that agonist movements of $\beta 1$ - $\beta 2$ relative to M2-M3 act as a brake on channel closing and hypothesized that increasing or decreasing the size of the α Val46-equivalent residue in GluCl should lead to corresponding changes in channel closing rates. In contrast to this hypothesis, the α Val46Ala mutation in the muscle nAChR has almost no effect on closing rates, yet dramatically influences channel function by reducing the rate of channel opening. These discrepancies may suggest that there are subtle differences in gating between homomeric and heteromeric pLGICs. As discussed in the text of the manuscript (lines 207 – 215), the effect of the α Val46Ala mutation on the rate of channel opening implies that there is tension at this interface in the apo state and that the reduction in volume of the side chain reduces this tension leading to a more stable closed conformation. This point is relevant to the discussion of the energies of closed and open states in point 2.

2. On page 6 (and other places), the authors state that “agonist-induced motions of the b1-b2 loop (and thus V46) release the constraints holding the M2-M3 loop in a closed conformation, which allows it to relax outward to open the channel”. This interpretation assumes that the closed pore configuration is higher energy or “tensed” and relaxes to the open channel when the b1-b2 loop moves away. The authors have no data on the thermodynamic stability or free energy of the open vs closed states of the pore and this interpretation remains purely speculative. In addition, the supposed stability of the open pore is in contradictions with free energy calculations done with GLIC by Zhu and Hummer [10.1073/pnas.1009313107]. To avoid confusion, I would avoid discussing the energetics whatsoever and replace “relax to an open conformation” with “form” or “adopt” throughout the manuscript.

We agree with the reviewer’s concern that we lack insight into the relative energies of the closed versus the open state of the TMD and thus must be cautious about interpreting our structural data in terms of “tensed” and “relaxed” conformations. In fact, we debated extensively this concept prior to submitting the work for publication. In this light, we have changed the title of the manuscript from:

“The relaxation of subunit conformational heterogeneity underlies allosteric transitions in a muscle nicotinic acetylcholine receptor”.

to

“A release of local subunit conformational heterogeneity underlies gating in a muscle nicotinic acetylcholine receptor”.

We have changed the wording throughout the manuscript, as shown in the marked-up version, to both soften the conclusions and to distinguish between fact and conjecture.

Despite this limitation, we still feel that the available data strongly supports a discussion of our results in the context of a release of conformational restraints model for several reasons.

First, although the reviewer notes that simulation data performed by Zhu and Hummer suggest that the closed state of the GLIC TMD is more stable than the open state, the authors of this work acknowledge that there are “considerable uncertainties” in their calculations. The interpretations of the Zhu and Hummer also need to be evaluated in the context that their energy calculations were based on the isolated open and closed TMD structures obtained from the open crystal structure of GLIC and modeled based on a closed structure of the homolog, ELIC. It has been argued, however, that the closed structure

of ELIC does not represent a resting state, but instead a distinct inactivatable conformation (Gonzalez-Gutierrez and Grosman (J. Mol. Biol., 2010 403, 693–705). The findings of this study must be interpreted in light of these concerns.

Second, Zhu and Hummer note explicitly that theirs is a “model” and they state that “the validity of our model remains to be tested in experiments”. They also state that they “predict in particular that a lipid membrane should show no increase in ion permeability after incorporating GLIC mutants with the ECD deleted, without observable single-channel currents”. This prediction, however, is not borne out in studies of isolated TMDs of the $\alpha 4\beta 2$ nicotinic and the $\alpha 1$ glycine receptor incorporated into liposomes, which exhibit substantial ion flux (references in the manuscript). In fact, Mowry et al. conclude from their study that “In the absence of the ECD, however, either the open-channel (conducting) conformation has become preferred or the energy barrier between open- and closed-channel conformations has been relaxed to allow spontaneous channel opening”.

Third, our data suggest that the principal α subunits impose a global closed phenotype onto the entire TMD. As also noted in the text, “this interpretation is strongly supported by a recent study showing that a non-agonist binding muscle type ancestral β subunit (β_{Anc}) forms homo-pentamers that undergo spontaneous bursts of channel openings²⁷, while the incorporation of a muscle α subunit represses these spontaneous channel openings leading to a closed, yet agonist activatable channel²⁸”.

In our opinion, points two and three are compelling and thus require that we discuss our data in the context of a “release of conformational constraints” framework. Our study provides new insight into how such restraints are imposed on the TMD by the ECD.

Regardless, we have acknowledged the lack of energetic calculations in the Discussion (lines 330 – 340 and 351-363). Again, we have been careful in the revised text to distinguish between facts obtained from our data and speculations derived from these observations.

3. Concerning pre-activation of the non-alpha subunits, it is stated on page 8 that “each non-alpha subunit adopts a locally active-like conformation at its intrasubunit ECD/TMD interface even prior to agonist binding”. How local are these changes? An interesting hypothesis would be that the asymmetry of the preactivated non-alpha subunits extends to the entire ECD rather than being limited to the interfacial loops. In this respect, it would be good to know if pre-activation at non-alpha subunits involves capping of the C-loop, beta-contraction/extension, global tilting and twisting of the ECD subunits, etc.

Thank you for this very important point, which we should have explored in the original manuscript. To assess the conformation of the ECD of each subunit in the apo and agonist-bound states, we used several parameters including those developed by Lev et al. (PNAS, 2017 114, E4158-E4167) and Calimet et al. (PNAS, 2013 110, E3987-E3996). The analysis presented here and in a revised Supplemental Information did not detect any parameter that suggests that the entire ECD of non- α subunits adopts a pre-activated conformation in the apo state.

To assess the capping of loop C, we first analyzed the apparent counterclockwise rotation of each subunit’s ECD that occurs during activation. To capture these changes, we aligned the TMDs of apo and bound states and calculated the ECD twist angle using the metric documented in Lev et al. Although

GLIC undergoes an $\sim 2^\circ$ decrease in twist angle between the closed and open states, there is essentially no change in twist angle for the α subunits of the nAChR (Figure 1, panel B). The non- α subunits also have little change in twist angle but appear to increase rather than decrease during gating. It appears that there are differences in the twisting motions between homomeric and heteromeric pLGICs.

We next quantified the ECD motions by measuring both the solvent accessible surface area (SASA) at the different ECD-ECD interfaces and the binding site contraction (Calimet et al.). The SASA of the subunit-subunit interfaces in the ECD decreased in all subunits (Figure 1, panel C), but the largest decreases occurred in both α subunits primarily due to a larger binding site contraction (Figure 1, panel D). Neither metric seem to provide any evidence for pre-activation elsewhere in the ECD of the non- α subunits. The loop C capping of the non- α subunits, which should be captured in the binding site contraction metric, did not provide evidence for pre-existence in either apo or bound states. This is likely because the non- α subunit C loops adopt unique conformations that are morphologically dissimilar to the α subunits (Figure 1, panel A) and are therefore difficult to directly compare.

Figure 1

We next used three previously defined metrics of activation, upper and lower ECD spreading and β -expansion, to assess whether the non- α subunits adopt a pre-active conformation. ECD spreading was calculated as the distance between the center of mass (COM) of the entire ECD at both the extracellular (upper) and TMD adjacent (lower) ends, to the COM of each subunit at the upper and lower ends. To find the COMs at the upper and lower ends of the entire ECD and the individual subunit ECD, an ellipsoid around the targets were created using the “measure inertia” command on ChimeraX using just the $C\alpha$'s of conserved β -strands in each subunit. The COM of the ellipsoid, along with the vector of the longest principal axis (corresponding to the height axis of the domain), were used to find COMs at two z positions at the top and bottom of each ECD (see panel A of the figure below). The distance between the full ECD COM and the subunit COM in both apo and bound states is defined as the ECD spread (Lev et al.) and is plotted on the right section of panel A of Figure 2. We found that there is a small inward motion of the upper ECD from each subunit, with no obvious change between a and non-a subunits. At the lower ECD, there does seem to be a subtle difference in spreading in a versus non-a subunits with a

Figure 2

larger spreading occurring in the non- α subunits. This is a product of the scissor-like motion that occurs during activation in the α subunits that is described in Zarkadas et al. (Neuron, 2022 110, 1358-1370). The magnitudes of the ECD spreads do not provide evidence for a pre-active ECD conformation of the non- α subunits.

We also used β -expansion, a parameter used for both GLIC (Lev et al.) and the homomeric $\rho 1$ GABA_AR (Cowgill et al. 2023, bioRxiv. Doi: 10.1101/2023.06.16.545288) to measure activation. We used the same protocol as previously described taking the COM of the α 's from two five residues stretches (one in the $\beta 1$ - $\beta 2$ loop and one in the $\beta 10$ -M1 linker) and measuring the distance in both apo and agonist bound state for each subunit (see panel B of the figure to the left). The expansion was originally described from the open to closed state of GLIC and thus is a contraction in the activation transition. We also see a subtle contraction in the α subunits,

but the non- α subunits instead expand. It is notable however that for both homomeric receptors, the magnitude of the contraction are $\sim 2\text{\AA}$, while here the magnitudes of the α subunit contractions are $< 0.2\text{\AA}$ and the non- α subunit expansions are $< 0.8\text{\AA}$. Furthermore, neither the contractions in the α subunits nor the expansions in non- α subunits suggest a pre-activate like conformation of the non- α subunits in the apo state.

Finally, we examined the changes in tilt angles that occur in each subunit ECD upon agonist binding. This was done in two ways: first by assessing how much each ECD changes its tilt angle along the z plane (relative to the pore) during activation and second by determining the relative change in tilt angle between the ECD and TMD in each subunit during activation. The former was measured by calculating

Figure 3

between the vectors of the longest principal axis of ellipsoids around the ECD and TMD of the same subunit for both apo and bound states. Panel C shows that the interdomain tilt angle in the non- α subunits increase during activation but decreases in the α subunits due to their scissor like interdomain motions.

Altogether, we were not able to identify any evidence for the pre-active like conformation of the ECD – TMD interfaces of the non- α subunit extending more globally to the remainder of the ECD. We state this explicitly in the revised manuscript (lines 251-254). We have also included this analysis in the Supplemental Information.

4. The mutagenesis studies point to a critical role of the aArg209 in interaction with the nearby anionic residues aGlu45, aAsp138, aGlu175. Since charge reversal or annihilation at Arg209 results in no expression, it is unclear at this stage whether this interaction involves receptor folding or gating. In addition, the location of these residues is reminiscent of the “switch of interactions” mechanism proposed in by Lev et al in GLIC [10.1073/pnas.1617567114]. In this paper, a switch of interaction between formally charged residues was proposed to mediate GLIC function by promoting tertiary changes at the lower beta-sandwiches, referred to as beta-contraction/expansion. Are similar changes

the change in angle between the vector derived from longest principal axis used to create the ellipsoids around each ECD and the z-axis (or pore axis). During activation, each subunit (other than the δ subunit) undergoes a straightening motion that decreases the angle relative to the pore axis (panels A and B in the figure to the left). This motion is conserved, and of a similar magnitude, in α and non- α subunits leaving no evidence for pre-activation of non- α subunits. It is notable however that while the non- α subunits seem to pivot with their membrane juxtaposed ends staying in the same position, the α subunits seem to translate toward the pore axis while also tilting (panel A). Because each subunit was aligned by their TMD, the relative translation of the α subunit ECD suggests the angle between ECD and TMD in the α subunits would differ from the non- α subunits. To measure this, we calculated the change in angle

observed in the muscle nAChR structures? Is there a switch of interactions involving Arg209? If so, this model should be discussed and the work of Lev et al mentioned.

Thank you for this important comment. We looked extensively at the salt bridges in the different subunits but we do not see any switch of interactions. The salt bridges involving the Arg209 equivalent in each subunit are relative stable upon agonist binding:

As already mentioned, we also looked at the possibility of a “ β sheet expansion” occurring in the nAChR models. Lev et al. report a “ β expansion” of $\sim 2\text{\AA}$ when going from the open to the closed conformation of GLIC. We performed the same analysis on the apo and agonist bound states of the nAChR and found that there were minimal and thus likely insignificant changes to the β -sheet compaction, with a contraction between 0.2\AA and 0.7\AA in the non- α subunits and an expansion that was less than 0.2\AA in the α subunits (see figure below).

Finally, the Lev et al. paper describe a how D32 (equivalent to E45 in the nAChR) in GLIC changes interaction partners from K248 on the M2-M3 loop in the closed state to R192 (equivalent to R209 in the nAChR) in the open state. Such a switch is not possible in the nAChR as there are no positively charged residues in the M2-M3 loops of any nAChR subunit.

5. The data in Fig.4C are puzzling. First, there is no panel C in the caption of Fig.4. Second, the large loss-of-function shown for α E45R in Fig.4C is inconsistent with the data reported in Table I (i.e., 15.6 gain-of-function). Third, the data shown in Fig.4C do not appear in Table I (i.e., α D138A, α E175A, etc.). Please correct/amend.

Thank you for noticing these errors. We have corrected Figure 4 to include a panel C label and have corrected the bar graph for the α E45R mutant. The data in Figure 4c is Table S6. We have noted this in the caption to improve clarity.

Minor points

- Page 6, line 254 “Mutations to the coordinated anionic residues were better tolerated than those of α Arg209, presumably because they can partially compensate for the loss of each other, but still lead to relatively large changes in EC50 values.”: Where does one find the data?
- Page 7, line 286: usually
- Page 8, line 351: “equivalent residues in principal agonist binding, but not principal agonist-binding subunits”. Unclear, consider revising.
- Page 9, line 369: include Ref to locally-closed structures of GLIC.

Fixed

Reviewer #1 (Remarks to the Author):

Given the time elapsed since the first review, I read the manuscript again as if for the first time. However, I came away with essentially the same impressions I had the first time, namely that much of what is presented was already known, but that the molecular level insight is new and fits well with what was known. In addition, I also took more time to study the tables and realized the authors did not use an important facet of the analysis that could have made their points stronger, and in one instance to come up with the opposite conclusion, as follows.

The inter-residue coupling analysis, as presented, resulted in the parameter ω , which is the ratio of the product of pairs of EC50 values (Tables 1 & 2). The authors correctly indicate that an ω value of one means no inter-residue interaction. However, values greater than or less than one contain information the authors fail to make use of. That is, take the natural logarithm of ω and multiply it by $-RT$ to yield an interaction free energy. In contrast to evaluating just ω , this places values greater or less than one on the same scale. In the case of the key interaction between Val46 and Pro265, the ω value of 0.15 converts to an interaction free energy of 1.14 kcal/mole, and that between Ile264 and Pro265 of 0.1 to 1.38 kcal/mole. Another ω value in Table 2 dealing with intra-subunit interactions was 0.01, which converts to 2.8 kcal/mole. This more complete analysis would allow the authors to make stronger inferences with more rigor.

In the case of inter-subunit interactions, the authors dismissed as unimportant the interaction between the M2-M3 loop of the alpha subunit and the F loop from the adjacent subunit gamma and delta subunits. For the mutant cycle aS266A/aT267A + gG182A/dG188A, an ω value of 0.05 was obtained, which converts to an interaction free energy of 1.79 kcal/mole, on par with the intra-subunit interaction energies. And this was for an interaction not previously known to be involved in gating. I expect the authors would like to reconsider their conclusion about the lack of inter-subunit interactions in gating.

Two further concerns have to do with style. Firstly, the authors apparently would like the work to be viewed as paradigm shifting. In one example, they claim the bevel gear model of interaction between a Val of the b1-b2 loop and a Pro of the M2-M3 loop is no longer valid. True the wrong Pro residue was used before, but this has been known for around a decade, and the correct Pro still contacts the b1-b2 Val residue, though in a slightly different orientation owing to the higher resolution available now. What is different, considering the author's new interpretation, is that rather than prying the channel open the bevel gear contact releases its constraint that keeps the pore closed. In another example, the transition from asymmetric to symmetric was known since the 1990s with Unwin's ACh-spray-freeze experiments at low resolution, which is confirmed by the author's release of conformational asymmetry shown with molecular level detail. Secondly, the key nuggets of discovery are not well highlighted, as they are buried in a large body of content repeating what is already known or of negative results. In short, the style is closer to that of a JBC paper for specialists rather than one for a general readership.

A last point is that while this release of conformational constraint mechanism likely applies to heteromeric cys-loop receptors, it may apply to homomeric receptors in some ways but not others. The intra-subunit contacts are likely common between the two classes of receptors, as the authors point out, but the asymmetric to symmetric transition may not. This may be obvious since homomeric receptors contain five identical subunits. However, there may be conformational asymmetry inherent to a pseudo-symmetric assembly of identical subunits that went undetected due to five fold averaging during structural modeling; multiple subunits may be in the primed conformation waiting to be released while others may already be released.

Reviewer #2 (Remarks to the Author):

The authors did a great job in replying to my comments/remarks and provided satisfactory answers and actions. My congratulations for a neat piece of work.

Reviewer #1 (Remarks to the Author):

Given the time elapsed since the first review, I read the manuscript again as if for the first time. However, I came away with essentially the same impressions I had the first time, namely that much of what is presented was already known, but that the molecular level insight is new and fits well with what was known. In addition, I also took more time to study the tables and realized the authors did not use an important facet of the analysis that could have made their points stronger, and in one instance to come up with the opposite conclusion, as follows.

The inter-residue coupling analysis, as presented, resulted in the parameter ω , which is the ratio of the product of pairs of EC_{50} values (Tables 1 & 2). The authors correctly indicate that an ω value of one means no inter-residue interaction. However, values greater than or less than one contain information the authors fail to make use of. That is, take the natural logarithm of ω and multiply it by $-RT$ to yield an interaction free energy. In contrast to evaluating just ω , this places values greater or less than one on the same scale. In the case of the key interaction between Val46 and Pro265, the ω value of 0.15 converts to an interaction free energy of 1.14 kcal/mole, and that between Ile264 and Pro265 of 0.1 to 1.38 kcal/mole. Another ω value in Table 2 dealing with intra-subunit interactions was 0.01, which converts to 2.8 kcal/mole. This more complete analysis would allow the authors to make stronger inferences with more rigor.

We agree with the reviewer that a quantitative assessment of the interaction free energy contributions to channel gating would be extremely insightful. As discussed in Fig. S1 of the Supplemental Information, however, although EC_{50} values depend on both binding (K_d) and gating equilibrium constant (θ), they are mathematically weighted towards the K_d so that an observed fold change in EC_{50} relative to WT is typically smaller than the calculated fold change in θ . Furthermore, the differences between the two values deviates further as θ deviates further from WT. For these two reasons, it is not possible to accurately calculate interaction free energy contributions to gating from our EC_{50} measurements.

We do appreciate that interaction energies add a linearity to the scale. For this reason, we have calculated “approximate” interaction free energies in Tables 2 and S3 but have noted that the values are only approximations provided for illustration purposes. Given the lack of rigour in such calculations, we are hesitant to incorporate these values into the discussion of our findings.

In the case of inter-subunit interactions, the authors dismissed as unimportant the interaction between the M2-M3 loop of the alpha subunit and the F loop from the adjacent subunit gamma and delta subunits. For the mutant cycle $\alpha S266A/\alpha T267A + \gamma G182A/\delta G188A$, an ω value of 0.05 was obtained, which converts to an interaction free energy of 1.79 kcal/mole, on par with the intra-subunit interaction energies. And this was for an interaction not previously known to be involved in gating. I expect the authors would like to reconsider their conclusion about the lack of inter-subunit interactions in gating.

We agree with the reviewer that we should have highlighted this interaction further. As discussed in the revised version, the simplest interpretation of both the energetic coupling and the structural data is that hydrogen bonds form between the polypeptide backbone C=O of $\gamma G182/\delta G188$ and the N-H of $\alpha S268$ in the agonist-bound state and thus contributes to activation. We have added additional data to the manuscript to highlight this interaction – specifically that the $\gamma G182A+(\alpha S266A+\alpha T267A)$ and

δ G188A+ α S266A+ α T267A) mutants are energetically coupled, but in contrast are independent on the S268P background. The current text in lines 167-181 now reads (with the new text underlined):

“Also, changing both γ Gly182/ δ Gly188 and α Ser266/ α Thr267 residues to alanine led to only a 3.4-fold loss of function despite an expected 66-fold loss of function if the individual mutants influenced function independently. This energetic coupling could be due to the disruption of a hydrogen bond that forms between the backbone carbonyl of γ Gly182/ δ Gly188 and the backbone amide group of α Ser268 that contributes energetically to the activated state (agonist binding decreases the hydrogen bond donor - acceptor distance from 3.4 Å to 2.9 Å; Fig. S3) as the losses of function observed for the γ G182A/ δ G188A mutations no longer affect function on the α S268P background (Table S4). Both the γ G182A and α S266A+ α T267A and the δ G188A and α S266A+ α T267A triple mutants affect function to a lesser extent than expected given the individual mutations. The mutations are also completely independent in the S268P background. On the other hand, the α S268P alone had no effect on the measured EC_{50} value. Although the structural and functional findings suggest that a backbone hydrogen bond between carbonyl of γ Gly182/ δ Gly188 and the amide of α Ser268 at the principal – complementary subunit interfaces contributes to activation, the limited energetic couplings across this interface suggest that interactions between side chains of γ/δ β 8- β 9/ β 10-M1 and $\alpha_\gamma/\alpha_\delta$ M2-M3 do not play a major role energetically driving the conformational changes that open the channel gate.”

We have added a Fig. S3 to highlight the formation of the hydrogen bond between the backbone carbonyl of γ Gly182/ δ Gly188 and the backbone amide group of α Ser268 upon agonist binding – Fig. S3. We have also revised the first paragraph of the Discussion in lines 310-314 to state:

“Although such interactions occur, such as the formation of a backbone hydrogen bond between the carbonyl of γ Gly182/ δ Gly188 and the amide group of α Ser268, the inability to identify consensus interactions across the ECD-TMD interface that define the open state, despite the extensive mutagenesis studies performed here and elsewhere^{15,16}, suggests that such open state defining interactions do not exist.”

Two further concerns have to do with style. Firstly, the authors apparently would like the work to be viewed as paradigm shifting. In one example, they claim the bevel gear model of interaction between a Val of the β 1- β 2 loop and a Pro of the M2-M3 loop is no longer valid. True the wrong Pro residue was used before, but this has been known for around a decade, and the correct Pro still contacts the β 1- β 2 Val residue, though in a slightly different orientation owing to the higher resolution available now. What is different, considering the author’s new interpretation, is that rather than prying the channel open the bevel gear contact releases its constraint that keeps the pore closed.

We agree that a major conclusion of our study is that instead of prying the channel open upon agonist binding, the ECD restrains the channel in the closed state and that agonist-induced allosteric communication involves a release of these conformational restraints. But we go beyond that to provide mechanistic insight at the residue level - insight that is critically important for a mechanistic interpretation. For example, although Lee and Sine (Nature (2005) **438**, 243–247) present compelling data that specific interactions between α E45/ α V46 on β 1- β 2 and α Val272 on M2-M3 are central to prying open the channel, the new structures show that β 1- β 2 does not contact α Val272 and that β 1- β 2 and M2-M3 move in roughly orthogonal directions upon agonist binding. The structures show that such a model is not correct. Xiu et al., (J Biol Chem (2005) 280, 41655-41666) proposed that the electrostatic pattern across the ECD-TMD interface governs allosteric communication, but no open state defining interactions were identified. In

contrast to Xiu et al., Cymes and Grosman (*PNAS* (2021) **118**, e2021016118) showed that specific residues at the ECD-TMD are not important for gating and that it is likely steric interactions at this interface that pry open the channel. The hypothesis of Cymes and Grosman, however, does not account for the compelling data of Lee and Sine, who show, for example, that the E45A+V46G double mutation leads to a ~6,000-fold reduction in the di-liganded gating equilibrium constant that almost abolishes the agonist-induced response. That latter suggest that specific residues are important in allosteric communication across the ECD–TMD interface.

It is at this residue level that the discrepancies between various models of allosteric communication at the ECD-TMD interface become apparent. Our findings present an alternative framework for understanding allosteric communication. We believe this framework better accounts for the available data in the literature.

In response to this critique, which was reiterated in both reviews, we have revised the manuscript to remove statements suggesting a paradigm shift in our understanding of allosteric communication in pLGICs and to present our findings with a neutral tone. For example, in the first submission, we stated in the Abstract:

“Here we use recent structures of the Torpedo acetylcholine receptor and screening mutagenesis to delineate the mechanisms underlying the coupling of agonist binding to channel gating.”

While in the current version we state:

“Here we use recent Torpedo acetylcholine receptor structures and functional measurements to delineate a key feature underlying allosteric communication...”

We have revised the text throughout the manuscript to highlight our findings as a “feature” of channel gating, as opposed to implying a huge paradigm shift. For example, we have revised one italicized heading, which stated:

“Channel opening is mediated by subunit asymmetric local structural rearrangements.”

to state:

“Channel opening is accompanied by subunit asymmetric local structural rearrangements.”

In another example, the transition from asymmetric to symmetric was known since the 1990s with Unwin’s ACh-spray-freeze experiments at low resolution, which is confirmed by the author’s release of conformational asymmetry shown with molecular level detail.

We agree with the reviewer that Unwin was the first to propose that there is subunit asymmetry in the resting state and that agonist binding relaxes this asymmetry to a symmetric state. We have cited this work in the Conclusion of each version of our manuscript.

We disagree, however, with the suggestion that the asymmetric to symmetric transitions highlighted in our work was known since the 1990s. Unwin et al. based their interpretations on a 4.6 Å resolution structure of the apo resting state and a 9 Å resolution structure of the activated state. Specifically, Unwin et al. proposed that the inner and outer β sheets in the α subunit ECD of the resting structure pack

differently than the inner and outer β sheets in the non- α subunits (β , γ , and δ), the latter adopting a packing similar to that observed in structures of AChBP. In the 9 Å resolution activated structure, no evidence for packing differences of the ECD β sheets could be identified. Unwin states (Unwin et al., J Mol Biol (2002) 319, 1165-1176):

“The core of the AChBP protomer is organized around two sets of β -stands, forming “Greek key” motifs, which are linked together through the Cys-loop disulphide bond and folded into a curled β -sandwich.¹⁵ This core was divided into separate “inner” and “outer” parts of the sandwich, which were fitted independently to the densities in the 4.6 Å, resting-state structure. We found that the two parts in all three non- α subunits (β , γ and δ) were oriented approximately as in AChBP, whereas the two parts in the α subunits had a different alignment. Although this distinction between the non- α and α subunits applied to the receptor in the resting state, no such distinction applied in the density map obtained from the activated receptor.”

We have performed an extensive analysis of the conformation of the ECD of each subunit to quantitatively assess whether the entire ECD of each non- α subunit adopts an activated-like state prior to agonist binding – we essentially tested the structural interpretations of Unwin et al. Using multiple parameters aimed at characterizing the conformation of each ECD (supplemental information pages 19 – 23), we found no evidence that the ECDs of the non- α subunits adopt an activated-like state prior to agonist binding. Our analysis thus contradicts the findings of Unwin et al. Furthermore, the low resolution of his structures did not allow him to make any inferences regarding the local asymmetry we characterize at the ECD – TMD interface. The findings we present in this manuscript are thus completely novel.

To address this concern, we have included in the “Supplemental Discussion of the conformation adopted by the ECD of each subunit in apo and agonist bound forms of the *Torpedo* nAChR” that the analysis was performed to test Unwin’s hypothesis. The first paragraph now states:

“To assess whether the local conformational asymmetry observed at the interface between the β 1- β 2 and M2-M3 loops in different subunits extends to the entire ECD of each subunit, as originally proposed by Unwin et al.⁵, we used several parameters including those developed by Lev et al.⁶ and Calimet et al.⁷”

Secondly, the key nuggets of discovery are not well highlighted, as they are buried in a large body of content repeating what is already known or of negative results. In short, the style is closer to that of a JBC paper for specialists rather than one for a general readership.

Although we agree with the reviewer, we have struggled with the issue of presenting the data in a conceptual context without discussing details of the experimental data. We have revised the Introduction and the Results sections to eliminate details and thus to focus on the main concepts in the paper. We have also revised the figures and Movie 1 to better illustrate the key issues explored in this work. Specifically, the new figures highlight the roughly orthogonal motions of the ECD and the TMD that occur at the intra-subunit interface, despite the extensive functional data showing that residues at this interface are key to gating. The figures and Movie 1 also highlight the concerted motions that occur between the ECD and TMD at the inter-subunit interface. The main theme of the work is to explore and thus understand the roles of the motions at these two interfaces in allosteric communication across the ECD-TMD interface.

Additionally, we have revised figures to make the presentation of structures more uniform. We have also added subsections to many figures so that each section of the text can more easily be related to a specific diagram to place the functional data in a visual context.

We hope that the revised text, figures, and movies will better convey the key concepts in the manuscript.

A last point is that while this release of conformational constraint mechanism likely applies to heteromeric cys-loop receptors, it may apply to homomeric receptors in some ways but not others. The intra-subunit contacts are likely common between the two classes of receptors, as the authors point out, but the asymmetric to symmetric transition may not. This may be obvious since homomeric receptors contain five identical subunits. However, there may be conformational asymmetry inherent to a pseudo-symmetric assembly of identical subunits that went undetected due to five fold averaging during structural modeling; multiple subunits may be in the primed conformation waiting to be released while others may already be released.

We agree and hope to see such structures solved in the future. To emphasize this point, we added the following statement to lines 397-398 in the Discussion:

“It remains to be determined whether subunit conformational asymmetry contributes to gating in homomeric pLGICs.”

Reviewer #2 (Remarks to the Author):

The authors did a great job in replying to my comments/remarks and provided satisfactory answers and actions. My congratulations for a neat piece of work.

We thank Reviewer #2 for their comments.

Reviewer #1 (Remarks to the Author):

Major concerns:

1. As pointed out previously, the organization of topics obscures the main point and confirmatory results are overemphasized. The main results of the paper supporting release of conformational constraint begins at line 199 of the main text, but this is preceded by largely confirmatory results (lines 90-110), reasoning of why macroscopic measurements of EC50 are less sensitive than single channel measurements of channel gating in documenting changes in channel function (lines 110-124), and description of the contributions of inter-subunit interactions to receptor activation (lines 125-198). It is hard to imagine a general reader making it through to the true essence of the paper.
2. Although inter-subunit interactions are investigated to the same extent as intra-subunit interactions, the former are largely dismissed as unimportant despite the inter-residue coupling energetics being similar between the two. For instance, the estimates of coupling energy between the alpha-M2-M3 linker and gamma/delta-beta8-beta9 linker residues (-2.4 to -7.4 kJ/mol) are on par with those between the alpha-M2-M3 linker and alpha-beta1-beta2 linker residues (-1.9 to -5.5 kJ/mol; Table 2). It is shown that the inter-subunit interaction does not change between apo and active structures, but this lack of structural change is also observed for the principal electrostatic interaction between alpha-pre-M1 and the alpha-beta1-beta2 linker. An objective reader could well be puzzled by this dismissal of inter-subunit coupling energies.

Other concerns:

1. Errors for the delta-delta-G values in Table 2 and elsewhere should be presented.
2. In Fig. S1, the mechanism should be presented (i.e. $A+R = AR = AR^*$ or $A+R = AR+A = A2R^*$).
4. The term bevel-gear is not a widely assumed mode of interaction between beta1-beta2 linker and M2-M3. Prior to the high resolution structures, all that was known is that the two structures articulated in some way to trigger channel opening. It seems fair to say that bevel-gear is one of several possible modes of interaction.
3. In Fig. S1 table, a theta value from the literature of 60 is assumed for wild type. This was for a mechanism in which a flip/prime state was the basis for the theta value. A more realistic value consistent with a model not including flip/prime would be around half this value. The table needs to be corrected accordingly. On this note, calculated theta values of 11,800 for Torpedo and 1610 for human receptors are unrealistic.

REVIEWER COMMENTS

We appreciate the reviewer's patience with our revisions and thank him/her for his/her comments. We believe these revisions have made the manuscript much stronger and better balanced.

Reviewer #1 (Remarks to the Author):

Major concerns:

1. As pointed out previously, the organization of topics obscures the main point and confirmatory results are overemphasized. The main results of the paper supporting release of conformational $\beta\beta\beta$ constraint begins at line 199 of the main text, but this is preceded by largely confirmatory results (lines 90-110), reasoning of why macroscopic measurements of EC50 are less sensitive than single channel measurements of channel gating in documenting changes in channel function (lines 110-124), and description of the contributions of inter-subunit interactions to receptor activation (lines 125-198). It is hard to imagine a general reader making it through to the true essence of the paper.

We agree with the reviewer that the discussion rationalizing why EC₅₀ values are less sensitive in documenting changes in channel function than single channel measurements is distracting from the main conclusions of the work. We have removed the entire discussion of this issue from the main text and have placed it in the revised Supplemental Fig. S1. We have referred to Fig. S1 at places in the manuscript where this discussion is relevant.

2. Although inter-subunit interactions are investigated to the same extent as intra-subunit interactions, the former are largely dismissed as unimportant despite the inter-residue coupling energetics being similar between the two. For instance, the estimates of coupling energy between the alpha-M2-M3 linker and gamma/delta-beta8-beta9 linker residues (-2.4 to -7.4 kJ/mol) are on par with those between the alpha-M2-M3 linker and alpha-beta1-beta2 linker residues (-1.9 to -5.5 kJ/mol; Table 2). It is shown that the inter-subunit interaction does not change between apo and active structures, but this lack of structural change is also observed for the principal electrostatic interaction between alpha-pre-M1 and the alpha-beta1-beta2 linker. An objective reader could well be puzzled by this dismissal of inter-subunit coupling energies.

We agree with the reviewer that this comparison is confusing. We have changed the language throughout the text to balance our discussion of the data obtained from mutations at the intra- and inter-subunit interfaces. For example, we no longer suggest that interactions at the intra-subunit interface are "critical" for channel function. We acknowledge that interactions at both interfaces are important.

In the revised manuscript, we focus the interpretations more on the α V46A mutation, which leads to a larger loss-of-function (particularly given the calculations in Fig. S1) than any mutation at the inter-subunit interface despite the fact that the mutation is a simple reduction in side chain volume. We also note that we detect numerous energetic couplings between α Val46 and residues on α M2-M3, while minimal energetic couplings are detected at the inter-subunit interface. We bolster the latter observations by explicitly discussing pertinent single channel measurements. Finally, we further highlight the remarkable observation that the simultaneous mutation of all four residues on α_7/α_8 M2-

M3 (α S266, α T267, α S268, and α S269), which structurally participate in inter-subunit interactions, to either Ala or Gly has almost no functional effect showing that interactions at this interface are not critical to channel gating. It is these comparisons that led us to the key structural observations presented in this work.

The track changes document highlights the extensive revisions that we have made to the text to address this concern.

Other concerns:

1. Errors for the delta-delta-G values in Table 2 and elsewhere should be presented.

Values have been included in the manuscript.

2. In Fig. S1, the mechanism should be presented (i.e. $A+R = AR = AR^*$ or $A+R = AR+A = A2R^*$).

The reaction mechanism has been included in Fig. S1

4. The term bevel-gear is not a widely assumed mode of interaction between beta1-beta2 linker and M2-M3. Prior to the high resolution structures, all that was known is that the two structures articulated in some way to trigger channel opening. It seems fair to say that bevel-gear is one of several possible modes of interaction.

We have removed the term bevel-gear from the text of the manuscript.

3. In Fig. S1 table, a theta value from the literature of 60 is assumed for wild type. This was for a mechanism in which a flip/prime state was the basis for the theta value. A more realistic value consistent with a model not including flip/prime would be around half this value. The table needs to be corrected accordingly. On this note, calculated theta values of 11,800 for Torpedo and 1610 for human receptors are unrealistic.

We originally estimated θ values from our EC_{50} measurements to highlight the non-linearity of the relationship between changes in EC_{50} values and changes in channel gating (θ). Reflecting on the reviewer's comments, we concluded that estimating θ values from our EC_{50} measurements, while illustrative, is fraught with too many assumptions. The estimated θ values thus lead to more confusion than clarity. We have thus retained the discussion of the general trends in Fig. S1 but have not calculated any θ values from our EC_{50} measurements. We feel that the discussion of trends conveys the key concepts to any reader who is interested in delving deeper into the data supporting the proposed mechanism and provides sufficient background to support our interpretations of the EC_{50} data.